# POLYRETINA restores light responses in vivo in blind Göttingen minipigs

Paola Vagni[1,3], Marta Jole Ildelfonsa Airaghi Leccardi [1,3], Charles-Henri Vila[1,3], Elodie Geneviève Zollinger[1], Golnaz Sherafatipour[1], Thomas J. Wolfensberger[2] & Diego Ghezzi [1✉]

Retinal prostheses hold the potential for artificial vision in blind people affected by incurable diseases of the outer retinal layer. Available technologies provide only a small field of view: a significant limitation for totally blind people. To overcome this problem, we recently proposed a large and high-density photovoltaic epiretinal device, known as POLYRETINA. Here, we report the in vivo assessment of POLYRETINA. First, we characterise a model of chemically-induced blindness in Göttingen minipigs. Then, we develop and test a minimally invasive injection procedure to insert the large epiretinal implant into the eye. Last, we show that POLYRETINA restores light-evoked cortical responses in blind animals at safe irradiance levels. These results indicate that POLYRETINA holds the potential for artificial vision in totally blind patients affected by retinitis pigmentosa.

[1] Medtronic Chair in Neuroengineering, Center for Neuroprosthetics and Institute of Bioengineering, School of Engineering, École Polytechnique Fédérale de Lausanne, Geneva, Switzerland. [2] Department of Ophthalmology, University of Lausanne, Hôpital Ophtalmique Jules-Gonin, Fondation Asile des Aveugles, Lausanne, Switzerland. [3] These authors contributed equally: Paola Vagni, Marta Jole Ildelfonsa Airaghi Leccardi, Charles-Henri Vila. ✉email: diego.ghezzi@epfl.ch

Retinal prostheses were introduced more than 30 years ago to restore vision in blind people affected by incurable diseases of the outer retinal layer[1,2]. So far, clinical trials with retinal prostheses have shown their potential in patients affected by retinitis pigmentosa[3,4] and age-related macular degeneration[5]. However, the devices tested in clinical trials are tiny and stimulate only a small portion of the retina. This limitation results in artificial vision produced only in a small part of the field of view: a significant limitation for totally blind people. Nevertheless, these results fostered novel concepts in neurotechnology, such as photovoltaic retinal implants[6–10]. A wireless photovoltaic device does not need implantable pulse generators, transocular connections and tracks in the electrode array, thus allowing the integration of thousands of electrodes at high density over a large surface[11]. Consequently, photovoltaic retinal prostheses can provide artificial vision over a wide visual angle by increasing the number of electrodes to enlarge the device's active area.

Several studies with sighted volunteers under simulated prosthetic vision highlighted the relevance of the visual angle during object recognition, mobility and navigation tasks[12–16]. The visual angle is proportional to the retinal coverage, while the stimulation resolution is linked to the electrode density. Therefore, increasing the electrodes' number, density and coverage are crucial for artificial vision in totally blind patients. To meet this goal, we recently developed a large and high-density epiretinal device (POLYRETINA)[11,17]. Results obtained ex vivo with retinas explanted from a mouse model of retinitis pigmentosa showed the ability to activate retinal ganglion cells (RGCs) at a safe irradiance level[18–20] with a high spatial resolution equivalent to the electrode pitch (120 μm)[17].

The next step is preclinical in vivo validation. However, testing POLYRETINA in an appropriate animal model brings on several challenges. The in vivo testing of biomedical devices is often performed in rodents. However, the small size of their eye does not allow the surgical placement of a large device like POLYRETINA. Moreover, for a photovoltaic retinal implant, the animal model must be insensitive to light to ensure that the recovered functions are directly linked to the activation of the prosthesis and not to the residual natural responses to light. Last, the surgical insertion of a large device requires a proper insertion tool to minimise surgical trauma and comply with standard surgical practice. We addressed all these issues and validated POLYRETINA using a functional in vivo assay in blind Göttingen minipigs.

## Results

**Development of a blind model in Göttingen minipigs.** Göttingen minipigs are widely used in ocular research[21–23] because they have a cone-rich part of the retina called area centralis[24,25]. Also, the eye size closely matches the one of humans[26], thus offering a good opportunity for the preclinical testing of large retinal prostheses[27].

However, the validation of photovoltaic retinal implants in Göttingen minipigs requires photoreceptor degeneration to decouple the prosthetic response from the natural response of the retina to light. Transgenic models of retinitis pigmentosa have been developed in miniature pigs[28,29], but their availability is limited outside the United States of America. An alternative approach relies on the intravenous injection of iodoacetic acid (IAA) to induce photoreceptor degeneration[30,31]. IAA interferes with the metabolism of photoreceptors by suppressing glycolysis in a concentration-dependent manner[32]. The effect of IAA has been documented in various animal models, notably in rabbits[33–36] and pigs[37,38]. In pigs, IAA leads to rod degeneration and cone inactivation[39]. However, there is a lack of information regarding the reaction to the toxin in the retina of Göttingen minipigs. Therefore, we developed and characterised a suitable animal model for translational research of photovoltaic retinal prostheses by the systemic injection of IAA at a dosage of 12.5 mg kg$^{-1}$, which was previously shown to induce maximal retinal degeneration in pigs[38]. The anatomical and physiological alterations caused by IAA in Göttingen minipigs were assessed using in vivo spectral-domain optical coherence tomography (SD-OCT), in vivo recordings of flash evoked electroretinograms (fERGs) and flash visual evoked cortical potentials (fVEP), and postmortem histological assays up to 3 months after IAA administration (see Table. 1 for the list of minipigs used in the study).

We monitored retinal degeneration over time by acquiring fundus (Fig. 1a and Supplementary Fig. 1a–e) and SD-OCT (Fig. 1b, c and Supplementary Fig. 1f–j) images. In a longitudinal experiment on one eye of an IAA-treated minipig and one eye of an untreated minipig, we quantified the thickness of the retinal layers at three distances (2, 5 and 8 mm) dorsal to the optic disc (Supplementary Fig. 1a) and observed anatomical changes induced by IAA. In the treated eye, the thickness of the outer retina appeared reduced by 40.81% (±6.56%, mean ± s.d. of the three distances) already from the first time point (1 month after IAA administration). On the other hand, the thickness of the inner retina slightly increased by 14.80% (±6.53%, mean ± s.d. of the three distances), while the total retinal thickness was overall reduced by 20.23% (±4.96%, mean ± s.d. of the three distances). These changes remained stable until 3 months after IAA administration (Supplementary Fig. 2d–f in black). Conversely, the untreated eye showed a constant retinal thickness over time (Supplementary Fig. 2d–f in grey). Based on the evidence that the outer retina already degenerates 1 month after IAA

**Table 1 List of Göttingen minipigs used in the study.**

| MP | IAA | H&E | IHC | SD-OCT | Echography | fERG | fVEP | fEEP |
|----|-----|-----|-----|--------|------------|------|------|------|
| 1 | no | 1e, f; S4, 5 | 1g, l; S6–17; 8g, h | S3 | | 2 | 3 | |
| 2 | no | | | S2, 3 | | 2 | 3 | |
| 3 | yes | S4, 5 | S6–17 | | | | | |
| 4 | yes | 1e, f; S4, 5 | | S3 | | 2 | 3 | |
| 5 | yes | S4, 5 | | S3 | | 2 | 3 | |
| 6 | yes | | 1g, l; S6–17 | S3 | | 2 | 3 | |
| 7 | yes | | S6–17 | S3 | | 2 | 3 | |
| 8 | yes | | S6–17 | 1a–c; S1–3 | | 2 | 3 | |
| 9 | yes | | 8 | | 5a; 6l | 7a, b | 7c–f | 7d, f |
| 10 | yes | | 8 | | 5a | 7a, b | 7c–f | 7d, f |
| 11 | yes | | 8 | | 5a | 7a, b | 7c–f | 7d, f |

Each cell contains the figure number where data are presented. S means supplementary. MP means minipig.

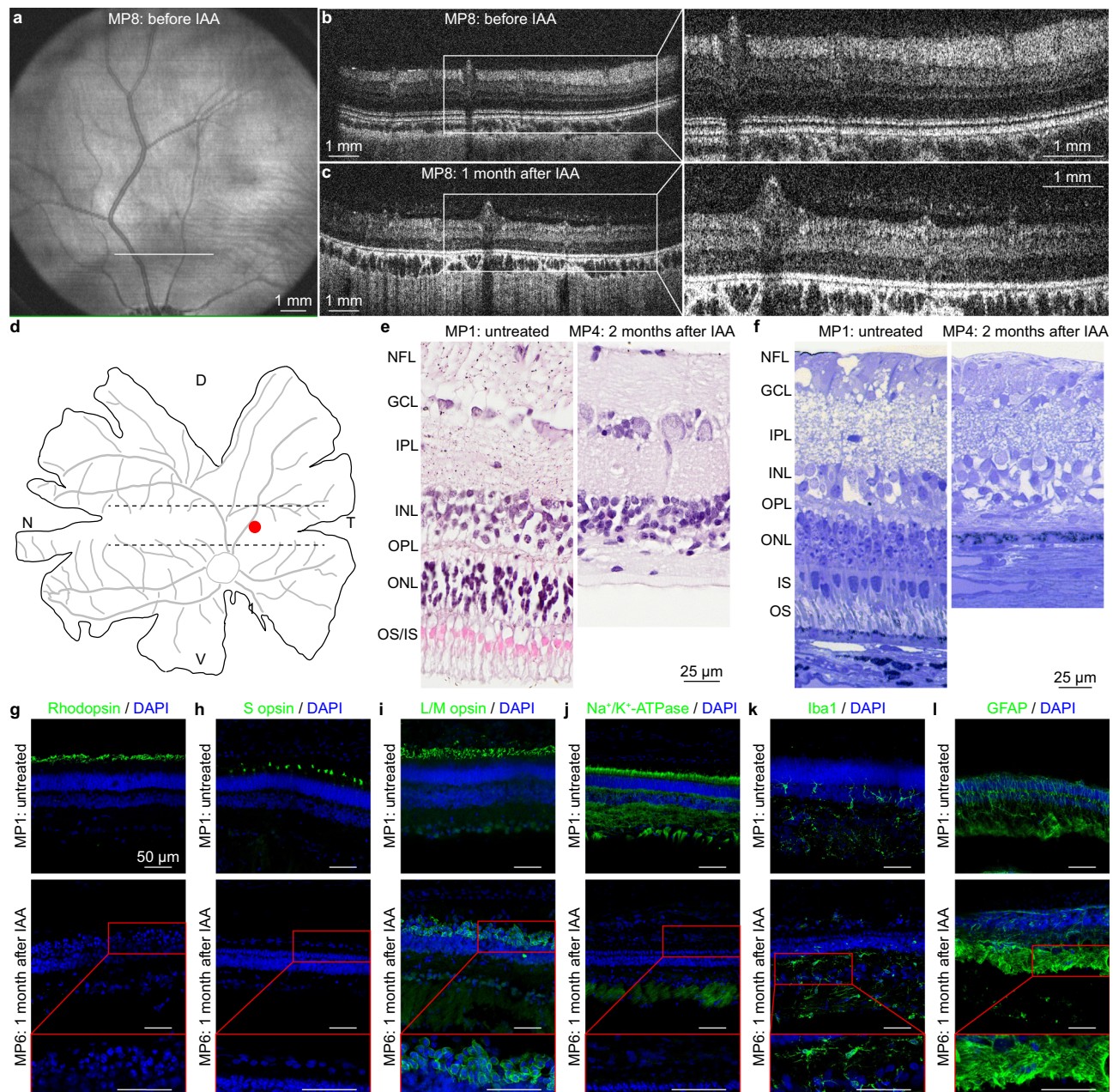

**Fig. 1 Retinal imaging in blind minipigs. a** Fundus image before IAA administration. **b**, **c** SD-OCT images of the retina before and 1 month after IAA administration, taken 2 mm above the optic disc (white line in panel **a**). The white boxes show a magnification of the retinal sections. Panels **a**–**c** are from MP8. **d**, Drawing of a flattened retina. The dashed lines delimit the area centralis, while the red circle indicates the point in the area centralis (central temporal) corresponding to the images in panels **e**–**l**. D dorsal; T temporal; V ventral; N nasal. **e** H&E staining of a retinal section in the area centralis (red circle in **d**) before and 2 months after IAA administration. NFL nerve fibre layer; GCL ganglion cell layer; IPL inner plexiform layer; INL inner nuclear layer; OPL outer plexiform layer; ONL outer nuclear layer; IS inner segments; OS outer segments. **f**, Semithin optical image of a retinal section in the area centralis (red circle in **d**) before and 2 months after IAA administration. Images in panels **e**, **f** are from MP1 and MP4. **g**–**l** IHC staining of a retinal section in the area centralis (red circle in **d**) against rhodopsin (**g**), S opsin (**h**), L/M opsin (**i**), Na$^+$/K$^+$-ATPase (**j**), Iba1 (**k**), and GFAP (**l**) before and 1 month after IAA administration. The red inserts show magnifications of the retinal sections 1 month after IAA administration. All scale bars in panels **g**–**l** are 50 µm. Images in panels **g**–**l** are from MP1 and MP6. H&E and IHC analyses have been performed at various time points ($n = 1$ eye from $N = 1$ minipig per time point).

administration, we performed a statistical comparison of the thickness of the different retinal layers in multiple treated eyes ($n = 10$ eyes from $N = 5$ minipigs) at this time point compared to the values before IAA administration. Results reported a statistically significant reduction of the thickness in the outer retina (Supplementary Fig. 3a, d, g in black), a statistically significant increase of the thickness in the inner retina (Supplementary Fig. 3b, e, h in black), and a statistically significant reduction of the thickness in the total retina (Supplementary Fig. 3c, f, i in black). The retinal thickness in multiple untreated eyes ($n = 4$ eyes from $N = 2$ minipigs) was not significantly altered at matching time points (Supplementary Fig. 3, in grey).

Postmortem histological assays confirmed the degeneration of the outer retina upon IAA administration. First, hematoxylin and eosin (H&E) staining of a retina 2 months after IAA

administration showed loss of the outer and inner segments of retinal photoreceptors, as well as degeneration of the outer nuclear layer (Fig. 1e) in the area centralis (Fig. 1d; red dot). The analysis has been performed at various time points ($n = 1$ eye from $N = 1$ minipig per time point) and eight locations corresponding to the peripheral nasal retina, the central nasal retina, the central temporal retina, and the peripheral temporal retina at the level of either the area centralis (Supplementary Fig. 4) or the optic disc (Supplementary Fig. 5). Two months after IAA administration, the outer retina appeared degenerated with no further gross anatomical changes occurring at the later time point tested (3 months after IAA administration). High-resolution optical images of semithin sections stained with toluidine blue confirmed the complete degeneration of photoreceptors in the area centralis (central temporal) 2 months after IAA administration (Fig. 1f).

Then, we performed immunohistochemistry (IHC) stainings to better analyse the retinal degeneration at various time points ($n = 1$ eye from $N = 1$ minipig per time point), up to 3 months after IAA administration (Fig. 1g–l and Supplementary Figs. 6–17). IHC staining against rhodopsin indicated complete degeneration of rods at every tested location at the level of both the area centralis (Fig. 1g and Supplementary Fig. 6) and the optic disc (Supplementary Fig. 7) for every time point. Similarly, IHC staining against S opsin showed degeneration of S-cones (Fig. 1h and Supplementary Figs. 8, 9). However, it is important to note that a small level of S opsin was detected in one minipig (MP6; 2.5 months time point), highlighting that a few S-cones might be spared, and the IAA treatment might have some variabilities among animals. IHC staining against L/M opsin showed longer protein preservation (Fig. 1i and Supplementary Figs. 10, 11). The minipig with spared S-cones also showed higher preservation of L/M opsin (MP6; 2.5 months time point). Nevertheless, despite the persistence of the cone opsins, we observed the degeneration of the cone outer segments, as indicated by the accumulation of S and L/M opsin in the cell body (Fig. 1i and Supplementary Figs. 8–11) caused by the inability to transport to the outer segments. The lack of the outer segment and the aberrant trafficking of the opsin are factors enhancing photoreceptor degeneration[39–41]. We further investigated the degeneration of the inner segments by IHC staining against the sodium/potassium-adenosine triphosphatase ($Na^+/K^+$-ATPase) enzyme. The IHC staining showed degeneration of inner segments (Fig. 1j and Supplementary Figs. 12, 13), with a few spared ones in one minipig (MP6; 2.5 months time point). Last, we evaluated IHC against two inflammatory markers: the ionised calcium-binding adaptor molecule-1 (Iba1) and the glial fibrillary acidic protein (GFAP). IHC against Iba1 (Fig. 1k and Supplementary Figs. 14, 15) and GFAP (Fig. 1l and Supplementary Figs. 16, 17) showed intra-animal and inter-animal variability but no notable difference among IAA-treated and untreated minipigs.

Next, we performed in vivo electrophysiology (fERGs and fVEPs) to confirm the light insensitivity of the retinas in IAA-treated minipigs upon presentation of 4-ms long white flashes delivered with a Ganzfeld stimulator. fERGs have been recorded with ERG-Jet™ electrodes in both dark-adapted (Fig. 2a–c) and light-adapted (Fig. 2d–f) conditions from both IAA-treated and untreated minipigs. Dark-adapted fERGs have been measured at increasing luminance levels (0.01, 0.05, 0.1, 0.5, 1, 5, 10 and 30 cd s m$^{-2}$), while light-adapted fERGs at 30 cd s m$^{-2}$ only. In both cases, five consecutive responses were averaged. The a-wave and b-wave of the fERG were suppressed in IAA-treated minipigs 1 month after administration in both dark-adapted (Fig. 2a, b) and light-adapted (Fig. 2d, e) conditions. Instead, the fERG response was preserved in untreated minipigs at matching time points in both dark-adapted (Fig. 2a–c) and light-adapted

(Fig. 2d–f) conditions. In dark-adapted conditions, the quantification of the a-wave and b-wave of the preserved fERG showed a logistic growth as a function of the luminance. In untreated minipigs ($n = 4$ eyes from $N = 2$ minipigs), the logistic growth is preserved at both time points for both waves (Fig. 2c). For the b-wave, the same logistic growth fits both datasets (before and after), indicating no statistical difference between the two time points. For the a-wave, the peaks after 1 month are slightly higher than before. Therefore, both datasets are fitted by logistic growth, but they are slightly statistically different. In IAA-treated minipigs ($n = 10$ eyes from $N = 5$ minipigs), the logistic growth fits the dataset only before treatment, while 1 month after IAA administration both a-wave and b-wave of the fERG are suppressed (Fig. 2b). The data before and after IAA administration are represented by two highly statistically different curves for both waves. Results in light-adapted conditions show a similar trend to those obtained in dark-adapted conditions. In untreated minipigs ($n = 4$ eyes from $N = 2$ minipigs), both a-wave and b-wave are preserved (Fig. 2f). For the b-wave, there is no statistical difference between the two time points. The a-wave is slightly higher after 1 month compared to before. Conversely, in IAA-treated minipigs ($n = 10$ eyes from $N = 5$ minipigs; Figs. 2e), 1 month after IAA administration both a-wave and b-wave of the fERG are suppressed. A previous report about IAA-treated pigs showed recovery of the cone response 5–6 weeks after IAA administration[39]. To rule out this possibility in the Göttingen minipig model, we performed a longitudinal study in both IAA-treated and untreated minipigs (Fig. 2g). Light-adapted fERGs showed suppression of the a-wave and b-wave in IAA-treated minipigs ($n = 7$ eyes from $N = 5$ minipigs) and no recovery for the entire testing period. On the other hand, they remained stable in untreated minipigs ($n = 4$ eyes from $N = 2$ minipigs).

Concurrently, we recorded fVEPs using cortical metal electrodes (Fig. 3a) in both dark-adapted (Fig. 3b, c) and light-adapted (Fig. 3d) conditions from both IAA-treated ($n = 10$ eye from $N = 5$ minipigs) and untreated ($n = 4$ eye from $N = 2$ minipigs) animals. Cortical electrodes were implanted as close as possible to the brain surface under radiography guidance. fVEPs were recorded from the metal electrode implanted in the hemisphere contralateral to the stimulated eye, and 30 consecutive responses were averaged. In both dark- and light-adapted conditions (Fig. 3c, d), fVEPs were suppressed in IAA-treated minipigs 1 month after IAA administration. Instead, fVEPs were preserved in untreated minipigs at matching time points. However, it is important to note that, in this experiment, cortical electrodes could not be left chronically implanted between recording sessions since the minipigs would have removed them, causing complications. Consequently, they had to be explanted and re-implanted in the same position before each recording session.

Collectively, these observations led us to conclude that IAA-treated Göttingen minipigs are a suitable blind animal model to evaluate the POLYRETINA prosthesis in vivo.

**Optimisation of POLYRETINA for in vivo testing in blind Göttingen minipigs.** Compared to the previous versions of POLYRETINA[11,17,19], we introduced some technical improvements. POLYRETINA is manufactured by plasma-bonding a photovoltaic interface (Fig. 4a–c) onto curved support (Supplementary Fig. 18a). Adjustments have been made to both structures.

SU-8 has been initially introduced in the photovoltaic interface as rigid protective platforms to prevent excessive mechanical stress on the photovoltaic pixels. The direct patterning by photolithography and the compatibility with the fabrication process made SU-8 an interesting material. However, an extra

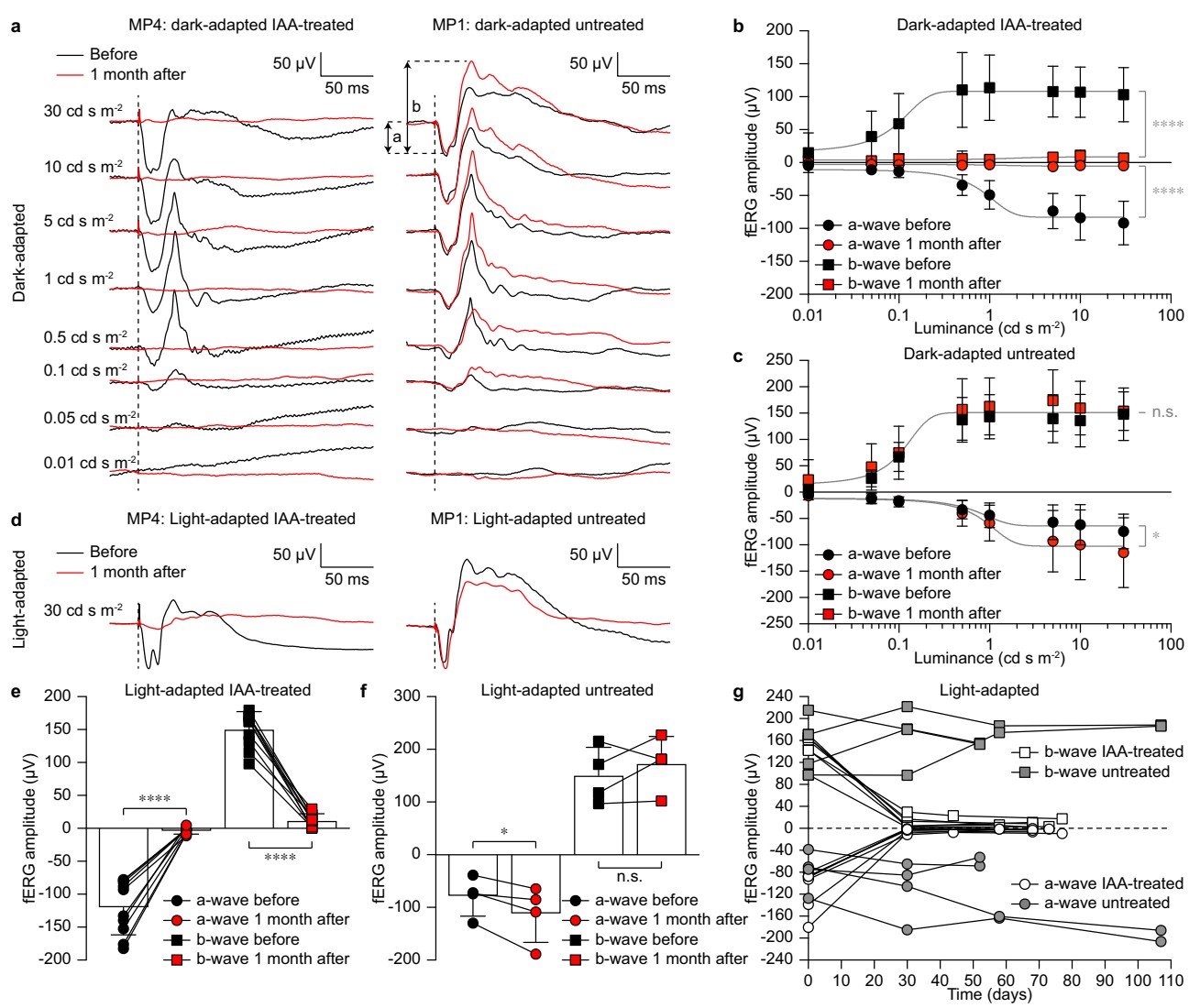

**Fig. 2 fERGs in blind minipigs. a** Dark-adapted responses in an IAA-treated minipig before and 1 month after IAA administration and in an untreated minipig at matching time points. The dashed lines indicate the flash onset. The quantification of the a-wave and b-wave is indicated. **b, c** Quantification of the dark-adapted fERGs in IAA-treated (mean ± s.d., $n = 10$ eyes from $N = 5$ minipigs) and untreated (mean ± s.d., $n = 4$ eyes from $N = 2$ minipigs) animals. Grey lines are the logistic growth regressions for IAA-treated (a-wave before: R2 = 0.6973; a-wave after: R2 = 0.0975; b-wave before: R2 = 0.3972; b-wave after: R2 = 0.0723) and untreated (a-wave before: R2 = 0.5283; a-wave after: R2 = 0.5368; b-wave before: R2 = 0.7283; b-wave after: R2 = 0.6258) minipigs. Extra sum-of-squares F test; IAA-treated: $p < 0.0001$ (****) for both a-wave and b-wave; untreated: $p = 0.0355$ (*) for a-wave and $p = 0.3235$ (n.s.) for b-wave. **d** Light-adapted responses in an IAA-treated minipig before and 1 month after IAA administration and in an untreated minipig at matching time points. The dashed lines indicate the onset of the flash. **e, f** Quantification of the light-adapted fERGs in IAA-treated ($n = 10$ eyes from $N = 5$ minipigs) and untreated ($n = 4$ eyes from $N = 2$ minipigs) animals. Two-tailed paired $t$-test; IAA-treated: $p < 0.0001$ (****) and power 1.00 for both a-wave and b-wave; untreated: $p = 0.0463$ (*) and power 0.97 for a-wave and $p = 0.4043$ (n.s.) and power 0.71 for b-wave. **g** Quantification of the light-adapted fERGs in IAA-treated ($n = 7$ eyes from $N = 5$ minipigs) and untreated ($n = 4$ eyes from $N = 2$ minipigs) minipigs over time. IAA was administered on day 0 after the recordings. Data in panels **a, d** are from MP4 (IAA-treated) and MP1 (untreated). Data in panels **b, c, e–g** are from MP4–8 (IAA-treated) and MP1–2 (untreated). Source data are provided as a Source Data file.

polydimethylsiloxane (PDMS) layer covering the platforms and homogenising the surface structure was necessary before the spin-coating of the poly(3,4-ethylenedioxythiophene) polystyrene sulfonate (PEDOT:PSS) layer and the bulk heterojunction composed by regioregular poly(3-hexylthiophene-2,5-diyl) and [6,6]-phenyl-C61-butyric acid methyl ester (P3HT:PC60BM). This extra PDMS layer reduced the maximum strain and handling resistance of the photovoltaic interface. Hence, in the new POLYRETINA, SU-8 has been replaced by parylene-C, a material with similar mechanical and optical properties, but deposited by chemical vapour deposition at room temperature with a homogeneous thickness. Parylene-C is patterned by dry

etching using a standard photoresist mask (Supplementary Fig. 18b). Therefore, POLYRETINA has been fabricated by consecutive deposition of parylene-C (5-μm thick) on top of a PDMS layer, spin-coating of PEDOT:PSS and P3HT:PC60BM, and sputtering of titanium coated with titanium nitride (Ti/TiN) at full-wafer scale without the need for a stencil mask, as in the previous version. Photolithography (possible on PDMS thanks to the parylene-C layer) and a series of dry etchings have been used to pattern Ti/TiN, P3HT:PC60BM, PEDOT:PSS and parylene-C, and to remove the residual photoresist. This process produced photovoltaic electrodes seated directly on parylene-C platforms (Fig. 4a, b), avoiding the strain from the PDMS directly

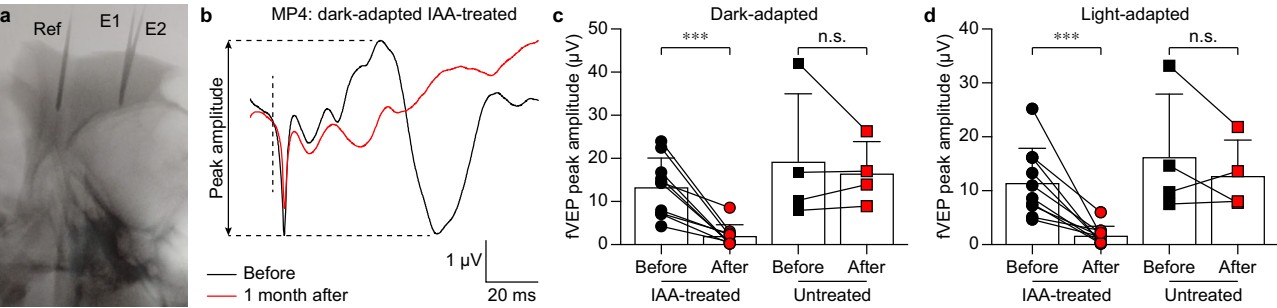

**Fig. 3 fVEPs in blind minipigs. a** Sagittal radiographic image of one minipig implanted with three electrodes for the recordings of fVEPs. Ref: reference electrode; E1: recording electrode 1; E2: recording electrode 2. **b** Dark-adapted fVEPs at 30 cd s m$^{-2}$ in an IAA-treated minipig before and 1 month after IAA administration. The vertical dashed line indicates the onset of the flash. The quantification of the peak amplitude is indicated. Data were from MP4 (IAA-treated). **c** Quantification of the dark-adapted fVEPs at 30 cd s m$^{-2}$ in IAA-treated minipigs (mean ± s.d.; $n = 10$ eyes from $N = 5$ minipigs) before and 1 month after IAA administration and in untreated minipigs (mean ± s.d.; $n = 4$ eyes from $N = 2$ minipigs) at matching time points. Two-tailed paired $t$-test; IAA-treated: $p = 0.0005$ (***) and power 1.00; untreated: $p = 0.5835$ (n.s.) and power 0.70. **d** Quantification of the light-adapted fVEPs at 30 cd s m$^{-2}$ in IAA-treated minipigs (mean ± s.d.; $n = 10$ eyes from $N = 5$ animals) before and 1 month after IAA administration and in untreated minipigs (mean ± s.d.; $n = 4$ eyes from $N = 2$ minipigs) at matching time points. Two-tailed paired $t$-test; IAA-treated: $p = 0.0008$ (***) and power 0.99; untreated: $p = 0.3900$ (n.s.) and power 0.71. Data in panels **c**, **d** are from MP4–8 (IAA-treated) and MP1–2 (untreated). Source data are provided as a Source Data file.

underneath the electrodes rising from the bonding (Supplementary Fig. 18a). Last, POLYRETINA electrodes were not encapsulated but left exposed since the parylene-C platform provided sufficient adhesion and protection.

The new electrode structure provided three advantages. First, the direct sputtering of Ti/TiN without a stencil mask allows for a higher degree of isotropy and homogeneous temperature distribution during deposition (Supplementary Fig. 18d, f). Hence, we observed a statistically significant increase in both the TiN surface roughness (1.033 ± 0.160 nm with a stencil and 2.850 ± 0.017 nm without a stencil; mean ± s.d., $n = 3$ samples; $p < 0.0001$, power 0.82, two-tailed unpaired $t$-test) and the TiN electrochemical surface area (percent increase compared to the geometrical surface area: 2.044 ± 0.190% with a stencil and 26.130 ± 1.802% without stencil; mean ± s.d., $n = 3$ samples; $p < 0.0001$, power 0.93, two-tailed unpaired $t$-test) measured with an atomic force microscope (AFM). Second, the photovoltaic electrodes had the same diameter as the parylene-C platform (100 μm; Supplementary Fig. 18b) and not 80 μm as in the previous version. A larger diameter increases the stimulation efficiency of the pixel since the photocurrent generated is proportional to the electrode area. Also, the larger electrode diameter decreased the optical transparency of the device to 31.48% ± 0.14% (mean ± s.d. of two POLYRETINAs) therefore reducing the total light reaching the retina. Third, the new structure allows for superior mechanical protection of the electrodes. During tensile testing, we observed that the photovoltaic interface could sustain up to 100% strain (Fig. 4d). At fracture, the photovoltaic electrodes were still intact as the fracture occurred between them. After plasma-bonding of the photovoltaic interface to the curved support (Supplementary Fig. 18a), POLYRETINA gained its curved shape (Fig. 4e), but still preserved high deformability, and the photovoltaic electrodes remained intact even during strong compressions and deformations (Fig. 4f, g).

The manufacturing process of POLYRETINA allows for the adaptation of the curved support to the real eye curvature, which, in Göttingen minipigs, was measured via non-invasive echography (Fig. 5a). Notably, the axial and lateral dimensions (18 and 22 mm, respectively) were preserved among all the different eyes measured in the study. The shape of the curved support was adjusted accordingly to match the eye curvature and ensure tight contact between POLYRETINA and the retina (Fig. 5b–g).

**Polyretina injection in blind Göttingen minipigs.** POLYRETINA folds tightly without mechanical damage to the

photovoltaic pixels (Fig. 4e–g): a necessary feature to insert the 14-mm wide implant through a 6.5-mm corneal cut. Inspired by intraocular lenses, we developed a surgical injector to insert the device into the eye (Supplementary Fig. 19). The injector is composed of three components: (1) a bevelled tube, (2) a narrow tube with thin and flexible extensions and (3) a plunger (Supplementary Fig. 19a). The bevelled tip facilitates the insertion through the corneal cut. During assembly, the narrow tube (2) is inserted into the bevelled tube (1), which lets the narrow tube slide freely (Supplementary Fig. 19b). The narrow tube (2) has two flexible extensions (0.35 mm × 1.7 mm × 16 mm) to accommodate the rolled POLYRETINA (Supplementary Fig. 19c). POLYRETINA is rolled with a pair of tweezers to achieve a maximum external diameter of 2.8 mm (Supplementary Fig. 19d, e) and placed between the two flexible extensions (Supplementary Fig. 19f). Upon retracting the narrow tube (2), POLYRETINA is loaded into the injector (Supplementary Fig. 19g). Last, the plunger (3) is inserted into the narrow tube (2) from the back (Supplementary Fig. 19g). The parallel extensions are designed to hold POLYRETINA and avoid friction while the narrow tube (2) slides forward into the eye. The plunger is used only during the last phase of the injection to push POLYRETINA out from the flexible extensions and release it into the eye avoiding damage to the eye structures. We performed injections in saline to verify the absence of any mechanical damage to the pixels after injection (Fig. 6a). The elasticity of POLYRETINA allowed it to unfold and recover its shape as soon as the flexible extensions opened up at the tip of the injector. A side-by-side comparison showed no difference in the photovoltaic pixels before (Fig. 6b) and after injection (Fig. 6c) as well as no overall sign of mechanical damage to the implant after injection (Fig. 6c).

After lensectomy and a 23 g vitrectomy with posterior vitreous detachment (Fig. 6d, e), POLYRETINA was successfully injected in blind Göttingen minipigs (Fig. 6g–i and Supplementary Movie 1) through a 6.5-mm long corneal incision (Fig. 6f). The epiretinal fixation was achieved using two custom-made stainless-steel retinal tacks (Fig. 6j, k). In the end, the corneal incision was sutured. Postoperative echography images showed that POLYRETINA matched the eye curvature with high adhesion to the retina (Fig. 6l).

**Functional validation of POLYRETINA in blind Göttingen minipigs.** We observed that the IAA treatment might have some variability among animals (Supplementary Figs. 8–13). Therefore,

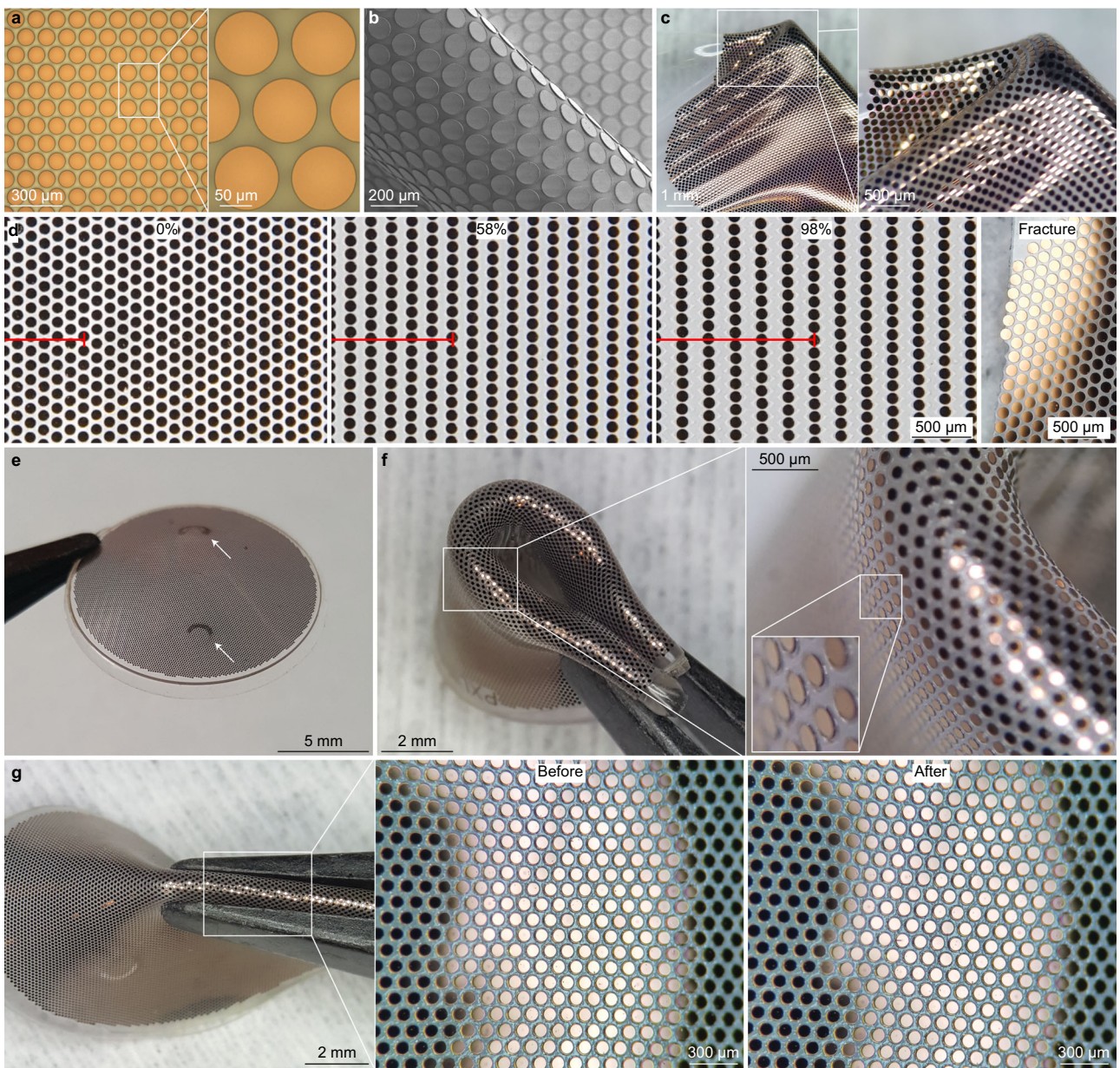

**Fig. 4 Optimisation of the POLYRETINA device. a** Picture of the photovoltaic pixels (100-μm diameter and 120-μm pitch). **b** Scanning electron microscopy picture of the photovoltaic pixels released from the wafer and folded to obtain tilted views on the pixels. **c** Image of a photovoltaic interface stretched and wrinkled with tweezers. **d** Pictures of the photovoltaic interface taken at 0, 54 and 98% of strain, and after fracture during a stretching test. The red bar shows six rows of pixels in the stretching direction. **e** Picture of a POLYRETINA. The white arrows indicate the points where tacks will be inserted. **f** Picture of a POLYRETINA folded four times. Images at various magnifications show that the photovoltaic pixels are intact during folding. **g** Picture of a POLYRETINA during pinching with magnifications on the intact photovoltaic pixels before and after pinching. Experiments in panels **d**, **f–g** have been reproduced in three replicates.

for the functional validation of POLYRETINA, we repeated IAA administration twice (3 weeks apart) and performed the surgical implantation and functional validation one month after the second dose. This time point has been chosen based on the previous electrophysiological assessment (Figs. 2, 3). Three minipigs were injected (Table. 1). The sample size was determined with a priori power analysis (alpha = 0.05, power 0.85). Lack of natural responses was assessed with fERGs and fVEPs evoked by 4-ms long white flashes delivered with a Ganzfeld stimulator. Responses two weeks before the surgery (which is 2 weeks after the second dose of IAA) were compared to recordings before IAA administration ($n = 6$ eyes from $N = 3$ minipigs). In the dark-adapted condition (Fig. 7a), the quantification of the a-wave and

b-wave of the fERG showed a logistic growth as a function of the luminance before IAA treatment. After IAA administration, both a-wave and b-wave are suppressed. The data before and after IAA administration are represented by two statistically different curves for both waves. Results in the light-adapted condition show a similar trend to the results obtained in the dark-adapted condition (Fig. 7b). After IAA administration both a-wave and b-wave are suppressed. Similarly, fVEPs were suppressed after IAA administration in both dark- and light-adapted conditions (Fig. 7c).

Blind Göttingen minipigs were unilaterally implanted in one eye for the functional validation of POLYRETINA ($n = 3$ eyes from $N = 3$ minipigs). For each animal, we selected for

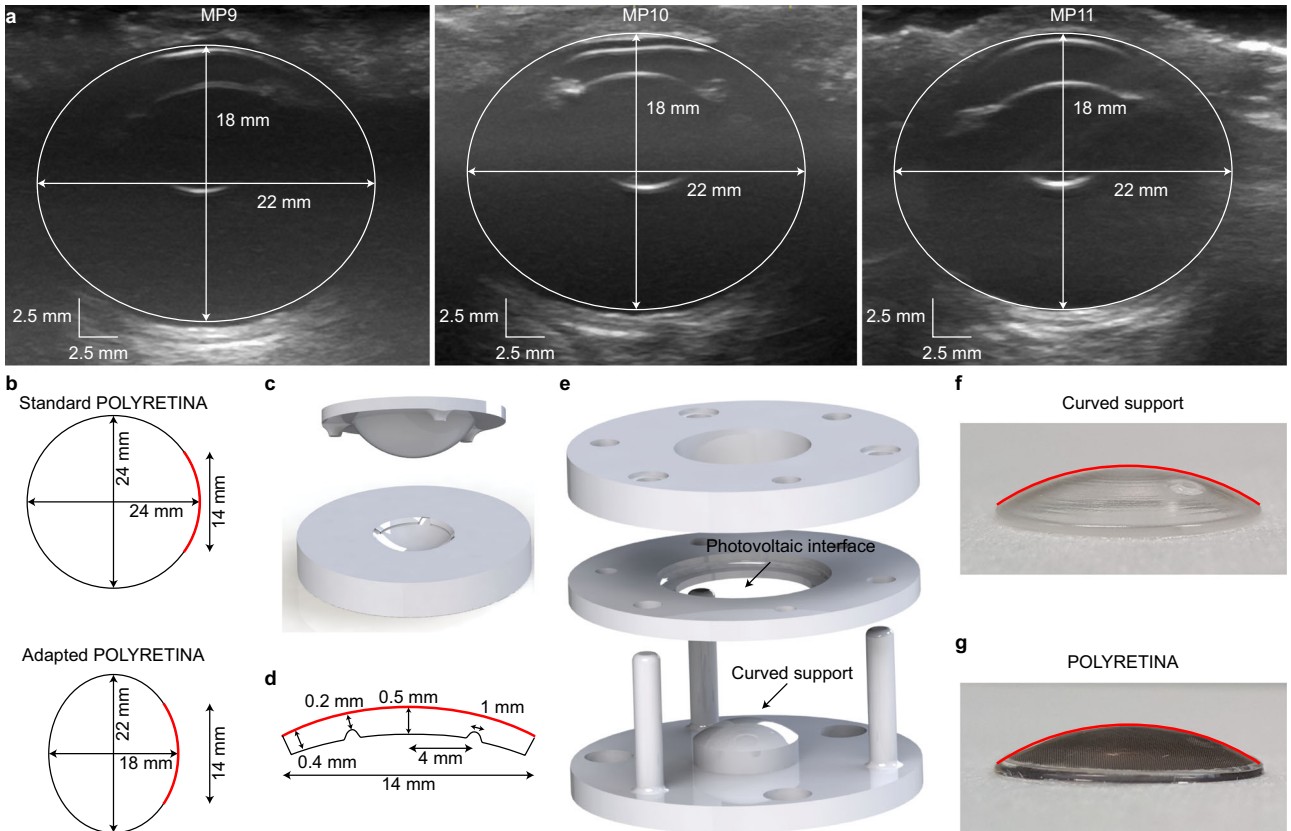

**Fig. 5 Adaptation of POLYRETINA to the eye of Göttingen minipigs. a** Echography of three minipig eyes. Images are from MP9–11. **b** Axial view of the standard POLYRETINA (top) and the POLYRETINA adapted to the ellipsoidal eye of Göttingen minipigs. The POLYRETINA is identified by the red line in both sketches. The black circle is the standard human eye (24 mm), and the black ellipse is the eye of Göttingen minipigs. **c** Three-dimensional model of the parts in poly(methyl methacrylate) used to produce the curved support by cast moulding. **d** Sketch of the cross-section of the curved support adapted to the minipig eye. The red lines correspond to the red line in **b**. The curved support has two indentations of 0.5 mm to identify the locations for the retinal tacks. **e** Three-dimensional model of the bonding setup. **f**, **g** Picture of the curved support (**f**) and the POLYRETINA (**g**) with the curvature of the curved support highlighted in red, showing that the curvature is not altered after bonding.

implantation the eye with the lowest peak-to-peak amplitude in the fVEPs. Then, we recorded flash electrical evoked potentials (fEEPs) to demonstrate the recovery of light responses in implanted blind Göttingen minipigs. Electrophysiological recordings were performed acutely before (fVEPs) and after (fEEP) POLYRETINA implantation, with the cortical electrodes left in place for paired comparison. Acute recordings were performed upon presentation of 10-ms long green flashes (565 nm) delivered with a collimated light-emitting diode. These parameters correspond to the optimal stimulation parameters suitable for the photovoltaic activation of POLYRETINA[17]. Recordings before surgery (i.e. before lensectomy and vitrectomy) showed no detectable fVEPs induced by the light pulse (Fig. 7d, black traces; averages of 200 consecutive responses) with peak-to-peak amplitudes (Fig. 7f, black circles) equivalent to the level of the peak-to-peak biological fluctuation measured in the same animal during a recording without light pulses (Fig. 7f, a grey filled area between dashed lines). After POLYRETINA implantation, fEEPs could be measured (Fig. 7d, red traces; averages of 200 consecutive responses) with peak-to-peak amplitudes (Fig. 7f, red circles) above the level of the peak-to-peak biological fluctuation (Fig. 7f, a grey filled area between dashed lines). Peak-to-peak amplitudes in fEEPs follow a semi-log trend as a function of the irradiance. The data before and after POLYRETINA implantation are represented by two statistically different curves. We also verified that fEEPs evoked by POLYRETINA persist over the simulation period of 200

responses (Fig. 7e). Response averages among the first half of the protocol (first 100 responses) and the second half of the protocol (second 100 responses) are qualitatively similar.

Our previously published results[11,17] identified 1 mW mm$^{-2}$ as the irradiance level within safety limits leading to a saturated response in mouse RGCs upon POLYRETINA activation. Therefore, we selected this value as standard irradiance for the functional activation of POLYRETINA. Here, we statistically compared the recovery of light sensitivity at the closest irradiance that we could obtain (1.18 mW mm$^{-2}$). The statistical analysis revealed that peak-to-peak amplitudes are significantly higher after implantation compared to before implantation ($p = 0.0074$, power $= 1.00$; two-tailed paired $t$-test). Also, the statistical analysis across all the irradiance levels showed that the first irradiance level leading to a statistically significant higher peak-to-peak amplitude is 33 µW mm$^{-2}$ ($p = 0.0308$, power $= 1.00$; two-tailed paired $t$-test). This value is coherent with the activation threshold we previously measured for full-field stimulation in retinal explants from blind mice (47.35 µW mm$^{-2}$)[11].

The presence of fEEPs after POLYRETINA implantation in completely blind minipigs (lacking fERGs and fVEPs before surgery) indicates the recovery of light sensitivity and links it to the POLYRETINA activation.

To further confirm that the cortical responses we recorded were induced by POLYRETINA activation and to rule out any contribution from surviving photoreceptors, we performed IHC on the three implanted eyes and verified the anatomical

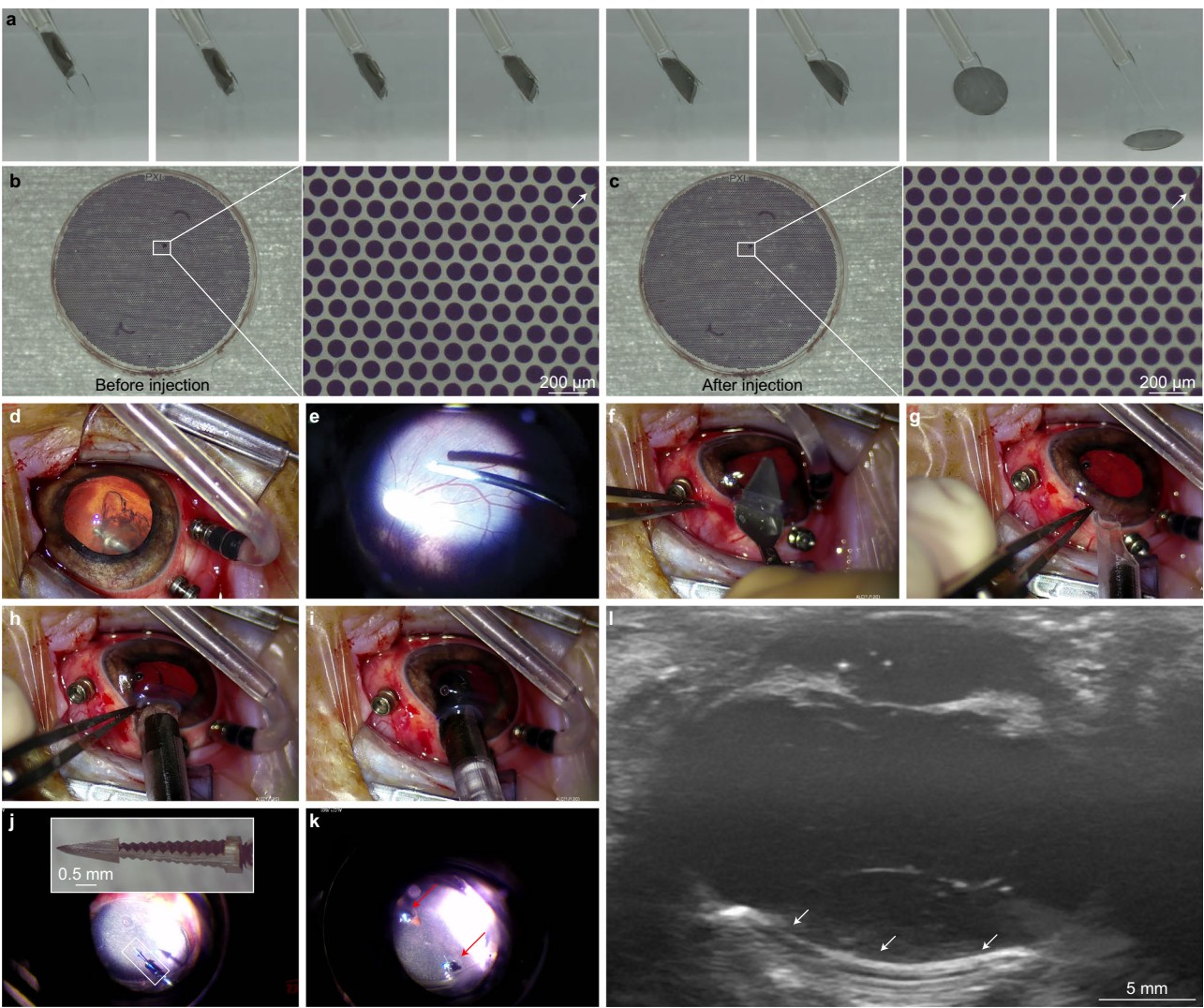

**Fig. 6 Surgical injection of POLYRETINA. a** Sequence of pictures showing POLYRETINA injection in saline (0.9% NaCl). When the implant is pushed towards the edge of the bevelled tube, the unfolding of the PDMS-based dome displaces the thin extensions that relieve the prosthesis. **b, c** POLYRETINA image before (**b**) and after (**c**) the injection procedure. The white arrows in the magnified views indicate a mark on the pixel to confirm that the image is from the same area. The experiment was reproduced in three replicates. **d–k**, Sequence of pictures showing the main steps of the injection of POLYRETINA in a blind Göttingen minipig eye: vitrectomy (**d**, **e**), a corneal incision (**f**), POLYRETINA injection (**g–i**) and POLYRETINA fixation with two retinal tacks (**j**, **k**). **l** Post-surgical echography of the implanted POLYRETINA (white arrows) showing its tight apposition to the retina. The experiment was reproduced in three replicates. Images in panels **d–l** are from MP9.

degeneration of photoreceptors (staining against rhodopsin, S opsin, L/M opsin and Na$^+$/K$^+$-ATPase). The eyes were enucleated 2 weeks after implantation (6 weeks after the second dose of IAA). IHC against rhodopsin (Fig. 8a), S opsin (Fig. 8b), L/M opsin (Fig. 8c) and Na$^+$/K$^+$-ATPase (Fig. 8d) showed no residual staining indicating complete degeneration of both rods and cones in the three tested animals. We report representative images from the central temporal retina at the level of the visual streak since this is the area that was always covered by POLYRETINA. The complete degeneration of photoreceptors might be explained by the double IAA treatment. This evidence further supports the recovery of light sensitivity by POLYRETINA.

Last, we performed IHC stainings against Iba1 (Fig. 8e) and GFAP (Fig. 8f) to assess the short-term immune response induced by POLYRETINA (2 weeks after implantation). Representative images from the central temporal retina at the level of the visual streak showed comparable levels of expression of the two markers among the three tested animals. We also

performed a quantitative analysis of the expression level for Iba1 (Fig. 8g) and GFAP (Fig. 8h) by computing the fraction of the retinal area showing the fluorescent marker in the three implanted minipigs compared to the untreated minipig. For each minipig, data from images at eight locations have been averaged, corresponding to the peripheral nasal retina, the central nasal retina, the central temporal retina, and the peripheral temporal retina at the level of either the area centralis or the optic disc. As previously remarked, IHC against Iba1 and GFAP showed some intra-animal and inter-animal variability. Iba1 expression (Fig. 8g) is variable among the three implanted minipigs, with one implanted minipig (MP11) having a statistically significant higher expression compared to another implanted minipig (MP9) and to the untreated minipig (MP1, Fig. 1k). On the other hand, GFAP expression (Fig. 8h) is not different among the three implanted minipigs and is comparable to the level detected in the untreated minipig (MP1, Fig. 1l). These results show a low level of acute immune response compared to the untreated minipig, which is a promising sign in view of a chronic POLYRETINA implantation.

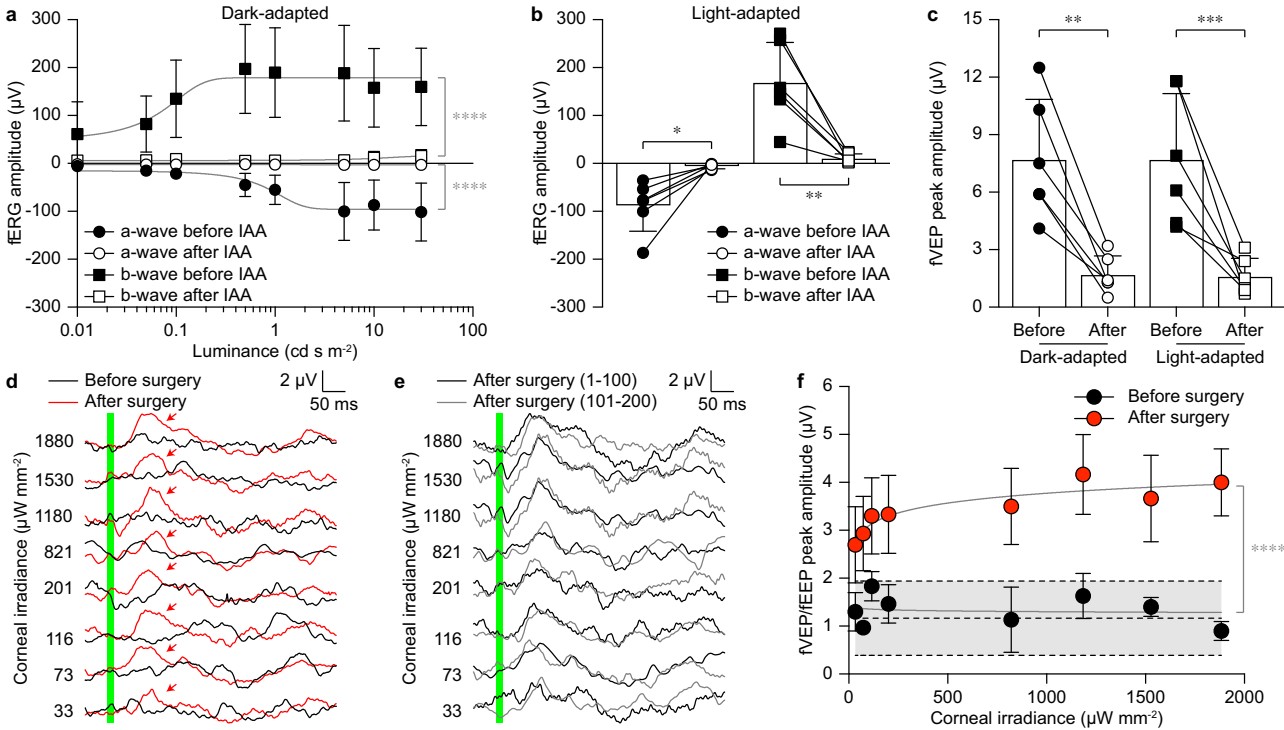

**Fig. 7 Functional validation of POLYRETINA in blind minipigs. a** Quantification of the dark-adapted fERGs in IAA-treated minipigs before and after IAA administration (mean ± s.d., $n = 6$ eyes from $N = 3$ minipigs). Grey lines are the logistic growth regressions (a-wave before: R2 = 0.49543; a-wave after: R2 = 0.0231; b-wave before: R2 = 0.2587; b-wave after: R2 = 0.1300). Extra sum-of-squares $F$ test; $p < 0.0001$ (****) for both a-wave and b-wave. **b** Quantification of the light-adapted fERGs before and after IAA administration (mean ± s.d., $n = 6$ eyes from $N = 3$ minipigs). Two-tailed paired $t$-test; a-wave: $p = 0.0138$ (*) and power 0.98; b-wave: $p < 0.0053$ (**) and power 1.00. **c** Quantification of the dark-adapted and light-adapted fVEPs in IAA-treated minipigs before and after IAA administration (mean ± s.d., $n = 6$ eyes from $N = 3$ minipigs). Two-tailed paired $t$-test; dark-adapted: $p = 0.0012$ (**) and power 1.00; light-adapted: $p = 0.0008$ (***) and power 0.99. **d** fVEPs and fEEPs were recorded immediately before (fVEPs, black) and immediately after (fEEPs, red) POLYRETINA implantation at increasing irradiance levels at the cornea. The green bar corresponds to the 10-ms long light pulse. Red arrows show the peak responses with POLYRETINA. **e** fEEPs recorded after POLYRETINA implantation were obtained from **d** by averaging two blocks of 100 consecutive responses. The green bar corresponds to the 10-ms long light pulse. **f** Quantification of the peak-to-peak amplitudes as a function of corneal irradiance before (black, fVEP) and after (red, fEEP) POLYRETINA implantation (mean ± s.d., $n = 3$ eyes from $N = 3$ minipigs). The grey area between dashed lines corresponds to the mean (±s.d.) biological noise obtained in each minipig from control trials without light stimulation. Grey lines are the semi-log regressions (before surgery: R2 = 0.0067; after surgery: R2 = 0.2847). Extra sum-of-squares $F$ test; $p < 0.0001$ (****). Data in panels **a–c**, **f** are from MP9–11. Data in panels **d**, **e** are from MP9. Source data are provided as a Source Data file.

## Discussion

POLYRETINA restores light sensitivity in vivo in blind Göttingen minipigs and shows good tolerability after two weeks of implantation. Ultimately, this preclinical validation of POLY-RETINA in a large blind animal model is an enabling step towards clinical validation.

First, we developed and characterised a model of chemically-induced blindness in Göttingen minipigs using IAA. Treated minipigs showed a substantial degeneration of the retinal outer nuclear layer, as evaluated using in vivo OCT and histological methods. Moreover, IAA treatment completely abolished both fERGs and fVEPs. The absence of light-induced electro-physiological responses makes the model suitable for testing photovoltaic retinal implants like POLYRETINA. IAA is a GAPDH (glyceraldehyde-3-phosphate dehydrogenase) inhibitor leading to rod degeneration and cone inactivation. Previous results showed that cone responses are rescued 5 to 6 weeks after IAA administration[39]. However, in our study, light-adapted fERGs were suppressed for the entire testing period (up to 11 weeks after IAA administration), indicating no recovery of cone responses. For POLYRETINA validation, we repeated IAA administration twice and compared the responses to light immediately before and after POLYRETINA implantation. Therefore, we exclude any possible contribution from remaining

or recovered photoreceptors. IHC stainings against Iba1 and GFAP did not show any remarkable inflammatory reaction upon IAA administration. This result is coherent with previous observations that IAA administration did not lead to any remo-delling of the surviving retinal cells[34], contrary to what is observed in genetic models of inherited retinal degeneration. On the other hand, IAA-treated minipigs do not recapitulate the complex remodelling processes undergoing in the retinas of patients affected by retinitis pigmentosa, which might hinder the translation of these results to patients.

Second, we developed and tested a minimally invasive injection procedure to insert POLYRETINA into the eye. Large retinal prostheses hold the potential of inducing artificial vision on a wide visual angle: a necessary feature to improve mobility skills, as well as faster and safer decision-making in blind recipients[16]. However, the insertion of large prostheses in the eye poses both technical and surgical challenges. To overcome those challenges, we draw inspiration from intraocular lenses that can be injected because of their deformability. We designed POLYRETINA as a stretchable prosthesis allowing deformation so as to be rolled, inserted through a small incision and self-open once released into the eye. A new injector device tailored to the mechanical properties of POLY-RETINA allows for safe injections and avoids damage to the device. Once injected, POLYRETINA recovers its original shape matching

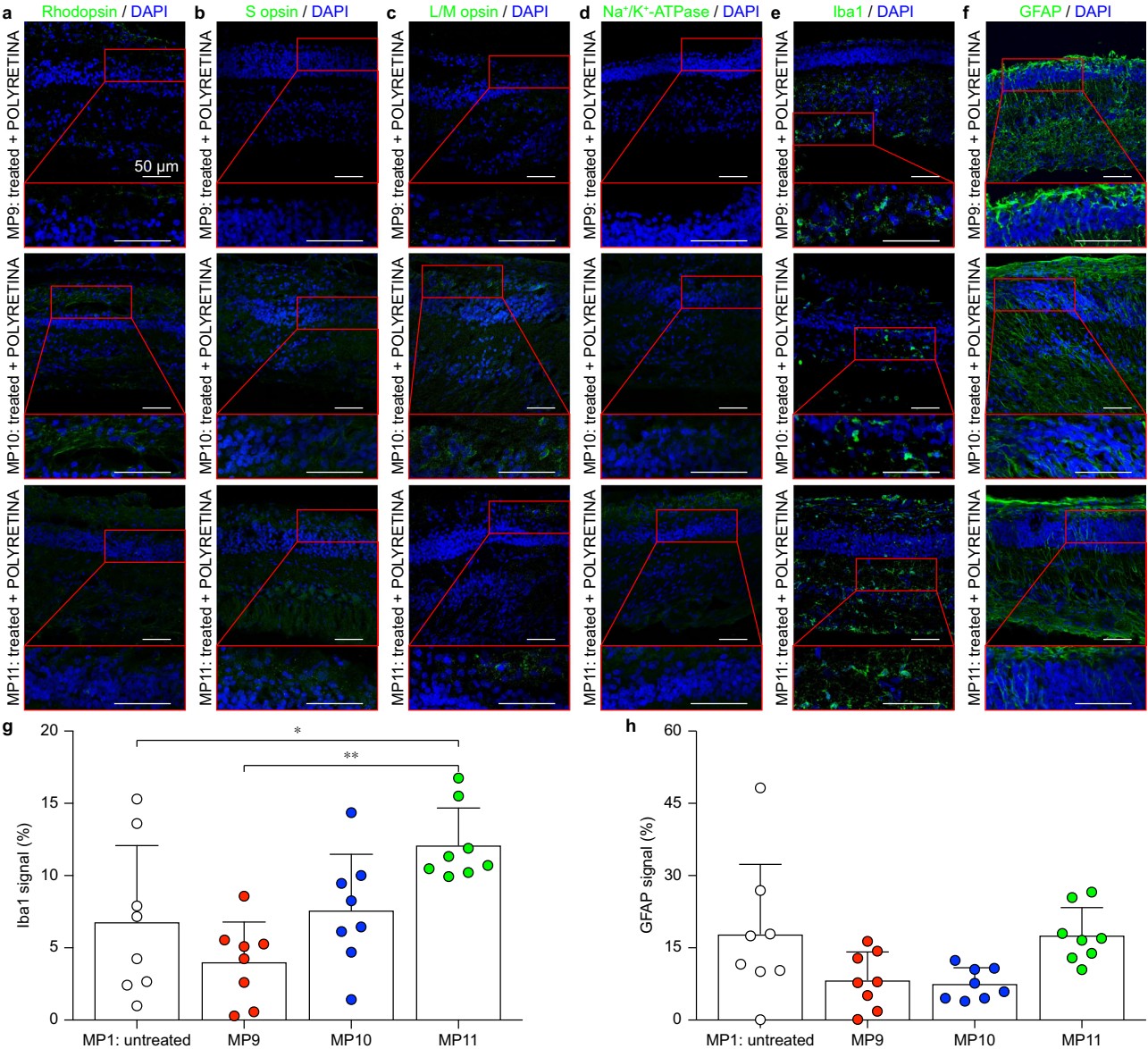

**Fig. 8 Histological assessment of implanted blind minipigs. a–f** IHC staining against rhodopsin (**a**), S opsin (**b**), L/M opsin (**c**), Na$^+$/K$^+$-ATPase (**d**), Iba1 (**e**) and GFAP (**f**) 2 weeks after POLYRETINA implantation. Each row corresponds to one implanted minipig. The red inserts show magnifications of the retinal sections. All scale bars in panels **a–f** are 50 μm. **g** Quantification of the expression level for Iba1 in the three implanted minipigs and one untreated minipig (mean ± s.d., $n = 8$ points from 1 eye for each minipig). One-way ANOVA: $p = 0.0022$ and $F = 6.263$; Tukey's multiple comparisons test: MP1 vs MP9 $p = 0.4756$, MP1 vs MP10 $p = 0.9732$, MP1 vs MP11 $p = 0.0422$ (*), MP9 vs MP10 $p = 0.2569$, MP9 vs MP11 $p = 0.0011$ (**), MP10 vs MP11 $p = 0.1048$. **h** Quantification of the expression level for GFAP in the three implanted minipigs and one untreated minipig (mean ± s.d., $n = 8$ points from 1 eye for each minipig). One-way ANOVA: $p = 0.0271$ and $F = 3.548$; Tukey's multiple comparisons test: MP1 vs MP9 $p = 0.1374$, MP1 vs MP10 $p = 0.0967$, MP1 vs MP11 $p > 0.9999$, MP9 vs MP10 $p = 0.9979$, MP9 vs MP11 $p = 0.1513$, MP10 vs MP11 $p = 0.1071$. In **g** and **h**, for each minipig, data from images at eight locations have been averaged, corresponding to the peripheral nasal retina, the central nasal retina, the central temporal retina, and the peripheral temporal retina at the level of either the area centralis of the optic disc. Images in panels **a–f** are from MP9–11. Data in panels **g**, **h** are from MP1,9–11. Source data are provided as a Source Data file.

the curvature of the implanted eye. This solution allows a tight apposition of POLYRETINA to the retina. POLYRETINA is held in position and fixed to the sclera using two retinal tacks, as it is conventionally conducted with epiretinal prostheses. Although retinal tacks have been used in surgical practice for many years, their use for retinal prostheses raises some concerns. In particular, the long-term mechanical stability of the implant and its anchoring to the retina might be affected. Also, it is difficult to control the pressure of the retina during surgery and in the follow-up period. Therefore, improvements in the fixation method are desirable.

Third, we reported the recovery of light perception by POLY-RETINA in blind Göttingen minipigs at safe irradiance levels. Three blind Göttingen minipigs were unilaterally implanted, and cortical potentials were recorded immediately before and immediately after POLYRETINA implantation. fVEPs were abolished by the administration of IAA, as demonstrated by recordings performed immediately before the surgery, but fEEPs were detected after POLYRETINA implantation, indicating the recovery of light perception. It is important to note that perception of light is just one step towards artificial vision. A key question is about spatial

resolution, which was not assessed in this work due to the challenges in recording cortical evoked potentials in minipigs. However, in a previous study performed ex vivo with retinal explants from blind mice, we reported that POLYRETINA holds a stimulation resolution equivalent to its pixel pitch, which is 120 μm[17].

In summary, the short-term in vivo preclinical validation of POLYRETINA showed safety and efficacy, thus representing an important step toward a first-in-men clinical trial. These results indicated that POLYRETINA holds the potential for artificial vision in totally blind patients affected by retinitis pigmentosa. Further studies will be required to assess the device in the long-term, addressing the functional and mechanical stability of the implant and its biocompatibility.

## Methods

**Ethical authorisation.** Experiments were approved by the Département de l'emploi, des affaires sociales et de la santé (DEAS), Direction générale de la santé de la République et Canton de Genève in Switzerland (authorisation number GE/120/19).

**Animal handling.** One-year-old female Göttingen minipigs (Ellegaard Göttingen Minipigs) were enroled in the study after 1 month of acclimatisation to their new environment. Minipigs were housed at least in pairs when possible, according to their origin dominance group to reduce stress and aggressiveness. Minipigs were fed a standard swine diet (SAFE127; SAFE Complete Care Competence) twice a day and received water ad libitum. The room temperature was set at 20 ± 4 °C with a 12/12 h light/dark cycle with a minimum of 40 lux and a maximum of 80 lux during the light period. Chains, pieces of wood and toys were used as environmental enrichment. No bedding materials were used to guarantee good sanitary conditions, but the environment was equipped with a rubber mat that was regularly washed. Minipigs had access to the outside and could freely move between the internal and external parts of the room. All the experiments were carried out during the day cycle. For the entire duration of the experiment, the general health condition was evaluated daily. Body weight was measured at least once per week. After any experimental procedure, the health condition was evaluated two times a day for one week. A summary of the minipigs used in the study is reported in Table. 1.

**Anaesthesia procedure.** Minipigs received a prophylactic antibiotic by intramuscular injection (enrofloxacin, Baytril 10%, 2.5 mg kg$^{-1}$). Antibiotic administration was repeated the two following days, once per day. Minipigs were premedicated with a mixture of azaperone (0.4 mg kg$^{-1}$), midazolam (0.75 mg kg$^{-1}$) and atropine (40 μg kg$^{-1}$) by intramuscular injection. Approximately 30 min after premedication, anaesthesia was induced by inhalation of sevoflurane (up to 6%), and an intravenous line was inserted in the ear vein. Atracurium (0.5 mg kg$^{-1}$) was administered intravenously to facilitate tracheal intubation. Minipigs also received an intravenous injection of the antibiotic cefuroxime (Labatec 1.5 g resuspended in 100 ml of NaCl 0.9%). After intubation, the sevoflurane was stopped, and the anaesthesia was maintained with continuous intravenous administration of propofol (8–10 mg kg$^{-1}$ h$^{-1}$) and ketamine (2 mg kg$^{-1}$ h$^{-1}$), while analgesia was assured via intravenous injection of fentanyl (2 μg kg$^{-1}$, 5–6 ml h$^{-1}$). Minipigs were constantly ventilated using a 30% oxygen fraction, with a tidal volume of 7 ml kg$^{-1}$ and a respiratory rate of 15 breaths per minute. Minipigs were placed on a heating pad to prevent hypothermia. Continuous monitoring of heart rate, electrocardiogram, temperature, blood pressure, end-tidal saturation and oxygen saturation was performed using a real-time anaesthesia monitoring system (Datex-Ohmeda). After the procedure, the anaesthesia perfusion was interrupted, and the oxygen fraction increased to 100%. The ventilator was set on pressure support to monitor the initiation of spontaneous ventilation. Upon giving signs of spontaneous ventilation, minipigs were extubated, and the respiration was assisted with a mask until signs of awakening were detected. Minipigs were then returned to their habitat and monitored until recovery. Analgesia was provided every 48 h by patches of buprenorphine (Transtec®, 35 μg h$^{-1}$) applied to the interscapular area.

**Iodoacetic acid injection.** IAA (I4386; Sigma-Aldrich) was dissolved in saline (NaCl 0.9%) at the concentration of 12.5 mg kg$^{-1}$ and injected intravenously under anaesthesia (5 ml in 15 min), followed by perfusion of 3 ml of saline (0.6 ml min$^{-1}$). The solution was prepared the same day of the procedure and kept on ice until the injection. Injections were performed at the end of the recording session.

**Spectral-domain optical coherence tomography.** Under anaesthesia, the pupils were dilated with atropine 0.5% (Théa Pharma) applied directly to the eye 30 min before starting the experiments. Images of the retina were obtained using an SD-OCT system (Envisu TM R2210 VHR; Bioptigen) with its proprietary software (InVivoVue, version 2.4.340; Bioptigen). Each B-scan image consisted of 100 to 1000 A-scans at a scanning rate of 32,000 scans s$^{-1}$. The covered area measured 12 mm × 12 mm, with a depth of 1.6492 mm (1000 × 200 × 1024 pixels). The optic

disc was positioned at the lower boundary of the image to ensure wide visibility of the visual streak. The superior dorsal margin of the optic disc was identified in the fundus image and defined as the 0-mm landmark. B-scans associated with the 2, 5 and 8 mm dorsal position relative to the landmark were imported in ImageJ (Fiji, version 1.53 m) and processed. The total retina thickness was defined from the edge of the retinal pigmented epithelium (identified as the edge of the first hyperreflective band) to the edge of the nerve fibre layer (identified as the edge of the most inner hyperreflective layer). The total retina was divided into two portions: the outer retina and the inner retina. The inner retina was defined as the sum of the nerve fibre layer, the ganglion cell layer and the inner plexiform layer. The outer retina was defined as the difference between the total and the inner retina. It contained the inner nuclear layer, the outer plexiform layer, the outer nuclear layer and the layer containing the outer and inner segments of photoreceptors. This method was chosen to perform a correct classification also in the degenerated retinas.

**Electrophysiology.** Under anaesthesia, the pupils were dilated with atropine 0.5% (Théa Pharma) applied directly to the eye 30 min before starting the experiments. fERGs were recorded with lens electrodes (ERG-Jet™, Fabrinal), using conductive gel to let them adhere to the eye. The responses were collected from the illuminated eye, while the contralateral eye was kept covered and used as a reference. fVEPs and fEEPs were recorded using two Kirschner wires of 1.6 mm in diameter (Medeco-ch) implanted in the skull in correspondence of the two visual cortices, while a third Kirschner wire was implanted more posterior as reference. The correct placement of the wires was monitored via real-time x-ray radiographic images during insertion. Kirschner wires were inserted as close as possible to the surface of the brain. ERG-Jet™ electrodes and Kirschner wires were connected to the amplifier (BM623; Biomedica Mangoni). One needle electrode was inserted in the ear and used as ground. The recordings were acquired simultaneously in two channels connected respectively to the ERG-Jet™ electrode placed on the illuminated eye and to the Kirschner wire placed on the contralateral visual cortex. The recorded signals were amplified, filtered (0.1–500 Hz), and digitised at 8 kHz (WinAver, version 1.12; Biomedica Mangoni). White light flashes (4-ms long) were delivered at 0.1 Hz repetition rate using a mini Ganzfeld stimulator (BM6007B 9.5; Biomedica Mangoni) positioned 1–2 cm from the eye. Green flashes (10-ms long) were delivered at a 1 Hz repetition rate using a light-emitting diode (565 nm, M565L3; ThorLabs) positioned at 15 cm from the eye and controlled with a programmable pulse train generator (Pulse Pal v2; Sanworks). Dark adaptation was performed under anaesthesia by leaving the animal in complete darkness for 30 min with the eyes covered with a black patch. Light adaptation was performed by exposing the eye to a continuous white light (10 min, 20 cd s m$^{-2}$) from the mini Ganzfeld stimulator. For chronic recordings, the Kirschner wires were removed before waking up the minipigs, and aluminium spray (Vetoquinol) was applied to protect the wound and facilitate healing. Data analysis was performed in WinAver and MATLAB (version 2020a; MathWorks).

**Immunohistochemistry.** Minipigs were euthanised while still under anaesthesia by intravenous injection of pentobarbital (Eskonarkon® 300 mg, 90 mg kg$^{-1}$), and the eyes were enucleated from the orbital cavity and placed in paraformaldehyde for 4 h. Eyes were cryoprotected in sucrose 10% for 6 h and then in sucrose 30% overnight. The eyecups were embedded in an optimal cutting temperature compound (Tissue-Tek®), frozen using a tissue snap freezing system (SnapFrost® 2; Excilone), and stored at −80 °C. 30-μm thick sections of the retina were obtained using a cryostat (CM3050S; Leica Microsystems) and placed on microscope slides. The sections were washed in phosphate-buffered saline (PBS), permeabilized with PBS + 0.1% triton (Sigma-Aldrich), blocked with blocking buffer (PBS + 0.1% Triton + 5% normal goat serum) and incubated with primary antibodies (1:300 anti-rhodopsin, Abcam AB5417; 1:500 anti-L/M opsin, Abcam AB5405; 1:500 anti-S opsin, Merck AB5407; 1:500 anti-Na$^+$/K$^+$ATPase (α3 Subunit), Sigma A273; 1:500 anti-Iba1, Wako 019-19741; 1:1000 anti-GFAP, Dako Z0334), overnight at 4 °C. The following day, the sections were incubated with secondary antibodies for 2 h (1:500 anti-mouse AlexaFluor 488, A11001, Thermofisher; 1:500 anti-rabbit AlexaFluor 488, SAB4600044, Sigma) and counterstained with DAPI 1:300 (Sigma-Aldrich). Finally, they were mounted with Fluoromount (Sigma-Aldrich) and imaged using a confocal microscope (LSM880, Zeiss) with its proprietary software (Zen, black edition; Zeiss). For all samples, representative images were acquired in four locations (peripheral nasal, central nasal, central temporal and peripheral temporal) at the level of both the area centralis and the optic disc (eight images per sample). Image quantification was performed in ImageJ (Fiji, version 1.53 m). First, a rectangular region of interest was manually selected to represent the area covered by the retina. Each image was converted to binary using a threshold. All pixels whose intensity was above the threshold were assigned the value 1, while the value 0 was assigned to the rest of the pixels. The threshold was set at 30 for GFAP. For Iba1, the threshold was defined using the Otsu method[42] in ImageJ (Fiji, version 1.53 m). The percentage of the pixels above the threshold was then computed. For each eye, the eight locations were averaged.

**Hematoxylin and eosin staining.** Enucleated eyes were placed in formalin 10% overnight. The eyes were then transferred into ethanol and prepared using an automated tissue processor (Tissue-Tek VIP® 6 AI; Sakura), where they were dehydrated by increasing concentrations of ethanol, followed by xylene, and finally embedded in paraffin. The paraffin-embedded samples were cut with a microtome

(HM355S; Thermo Scientific) at a thickness of 5 μm. Hematoxylin and eosin staining was performed on the sections using a multistainer (ST5020; Leica Microsystems). Images were acquired using a slide scanner microscope (VS120; Olympus) with its proprietary software (VS-ASW; Olympus).

**Semithin sections**. Enucleated eyes were placed in a solution of 3.2% paraformaldehyde and 1.25% glutaraldehyde in 0.1 M phosphate buffer and left for 4 h at room temperature. Strips of tissue (~2 mm$^2$ × 10 mm$^2$) were cut using a razor blade perpendicular to the plane of the retina and placed in the same fixative overnight at 5 °C. Each strip was vibratome sectioned at 200 μm thickness. Slices were then washed thoroughly with cacodylate buffer (0.1 M, pH 7.4), postfixed for 1 h in 1% osmium tetroxide with 1.5% potassium ferrocyanide, and then followed by 1 h in 1% osmium tetroxide alone. Sections were finally stained for 1 h in 1% uranyl acetate in water before being dehydrated through increasing concentrations of alcohol and then embedded in Durcupan ACM resin (Fluka). The resin was hardened at 60 °C for 24 h, and 0.5-μm thick sections were cut with a diamond knife and mounted onto glass slides. Sections were stained with toluidine blue, and images were collected with a light microscope using transmitted light.

**Polyretina manufacturing**. Photovoltaic interfaces were fabricated on silicon wafers. A thin sacrificial layer of PSS (561223; Sigma-Aldrich) was spin-coated on the wafers (1500 rpm, 60 s) and baked (145 °C, 10 min). Degassed PDMS pre-polymer (10:1 ratio base-to-curing agent, Sylgard 184; Dow-Corning) was then spin-coated (900 rpm, 60 s) and cured in the oven (75 °C, 2 h). After surface treatment with oxygen plasma (30 W, 30 s) and Silquest A-174NT silane, a 5-μm thick parylene-C layer was deposited via chemical vapour deposition (C25S; Comelec) by pyrolysing 10 g of Galxyl C precursor. PEDOT:PSS (Clevios PH1000; Heraeus) was mixed with 0.1 vol% (3-Glycidyloxypropyl)trimethoxysilane (440167; Sigma-Aldrich), filtered (polyethersulfone filters, 0.2-μm; Corning), and then spin-coated (3000 rpm, 40 s) onto the parylene-C surface previously treated with oxygen plasma (100 W, 30 s). Annealing was then performed (115 °C, 30 min). The preparation of the P3HT:PC60BM blend was performed in a glovebox under a nitrogen atmosphere: 20 mg of P3HT (M1011; Ossila) and 20 mg of PC60BM (M111; Ossila) were both dissolved in 1 ml of anhydrous chlorobenzene each, and left stirring overnight at 70 °C. The solutions were then filtered (polytetrafluoroethylene filters, 0.45 μm; Corning) and blended (1:1 v:v). The P3HT:PC60BM blend was then spin-coated at 1000 rpm for 45 s and annealed at 115 °C for 30 min still under a nitrogen atmosphere. Ti (100-nm thick) and TiN (100-nm thick) were deposited by direct current (400 W) and radio frequency (200 W) magnetron sputtering, respectively. Photolithography with an 8-μm thick photoresist was performed to pattern the photovoltaic pixels. Then, dry etching of Ti and TiN was obtained with a gas mixture of 20 sccm dichlorine, 30 sccm argon and 15 sccm helium (radio frequency power 50 W, inductively coupled plasma power 800 W, pressure 10 mTorr; Corial 210 IL); subsequently, oxygen plasma was applied to etch P3HT:PC60BM, PEDOT:PSS, parylene-C and the residual photoresist (radio frequency power 150 W, inductively coupled plasma power 500 W, 50 sccm oxygen and 17 sccm helium, 5 mTorr; Corial 210 IL). An endpoint detection system was used to determine the duration of the etch and stop it as soon as the layers were removed. The wafers were then placed in deionised water to allow the dissolution of the PSS sacrificial layer and the release of the photovoltaic interface. The floating membranes were collected and dried in the air. The curved support was fabricated using a milled poly(methyl methacrylate) mould, filled with PDMS pre-polymer (5:1), which was then degassed and cured in an oven (80 °C, 2 h). The photovoltaic interface was placed in its holder, clamped with an o-ring, and held with the top circular part. The curved support was placed on its support. After activation by exposure to oxygen plasma (20 W, 30 s; Diener ZEPTO) of both the photovoltaic interface and the curved support, they were contacted with a drop of uncured PDMS (5:1) to allow uniform bond thanks to radial stretching of the photovoltaic interface. The whole system was placed under load (1 kg) in an oven at 80 °C for at least 2 h, released after cooling, and the excessive PDMS used to clamp the array was removed by laser cut.

**Sterilisation procedures**. POLYRETINAs were sterilised in the oven at 80 °C for 4 h. Retinal tacks, tack holders and tweezers were dipped in ethanol (70%) and placed in the oven at 80 °C for 4 h. Injectors were sterilised in ozone, as they are sensitive to heating and solvents.

**Transmittance measurements**. Light transmittance of POLYRETINA was evaluated by using a light-emitting diode (M565L3, ThorLabs). Transmitted light was measured with a power metre (PD300-R Juno; Ophir Optronics Solutions). The light power was compared before and after placing POLYRETINA in the light path.

**Atomic force microscope and surface roughness**. AFM images were obtained with a Bruker Dimension icon microscope and ScanAsyst-air Si tips. Images (500 nm$^2$ × 500 nm$^2$) were plotted, and the root mean square roughness and the surface area were calculated with the analysis software (NanoScope, version 9.4; Bruker). The surface area was normalised to the nominal flat area (250,000 nm$^2$).

**Polyretina injection**. After anaesthesia, the minipigs were set in lateral decubitus position with the head slightly tilted so as to expose the eye to be implanted on a horizontal plane. The skin surrounding the eye was shaved, washed with soap and then dried. The eye was disinfected with drops of povidone-iodine 5% (1 ml). Betadine 5% was applied three times on the skin and eyelids and left to act for 2 min before being wiped off with sterile gauzes. A sterile field was prepared, leaving only the eye visible. The surgery was performed using a microscope (M822 F20; Leica Microsystems) with an OCULUS BIOM® ready. Surgical procedures were recorded with a camera system (EvoHD; Leica Microsystems). Surgical videos were edited in iMovie (version 10.3.2; Apple). Eyelid retractors were applied, and a small incision of 1–2 mm was performed on the lateral canthus so as to enlarge the access to the sclera. Viscoshield (VISCO SHIELD® HPMC Viscoelastic; Oasis medical) was applied regularly throughout the surgery to preserve the cornea. An incision using a Pic 23 G Angled (MANI) was performed on the limbus to allow the placement of an irrigation tube connected to an Alcon BSS + infusion in order to maintain the ocular pressure throughout the surgery in the range of 30 to 40 mmHg. Two other incisions were performed, allowing the insertion of two cannulas (Caliburn 23 G Cannulas 1-step, self-sealing), enabling easy access of 23 G instruments to the orbit. A lensectomy was performed followed by a 23 G pars plana vitrectomy (Associate® 2500 Compact System; Dutch Ophthalmic) to allow intraocular manipulations and insertion of the prosthesis. Triamcinolone (Vitreal S; Horus Pharma) was used to stain and visualise membranes and remaining vitreous, ensuring its complete removal and that the prosthesis would be in close contact with the retina. In the eventuality where the iris would contract, the preparation of the eye was concluded by applying disposable iris retractors leaving the posterior chamber easily accessible. Measurements were taken using a calliper on the cornea to prepare for a 6.5 mm tunnel. A Scalpel (MANI 11°) was used to create an initial grove. An angled crescent knife was then used to perform a corneal tunnel which was further enlarged by diamond-shaped slit knives with increasing widths from 3.2 to 4 to 5.5 mm and finally 6 mm (MANI). The custom injector containing the rolled prosthesis was inserted into the incision until the entire cross-section of the bevelled tube entered the eye chamber. The narrow tube was then pushed completely through the cannula, allowing the flexible wings to naturally open, initiating the unrolling of the prosthesis and its release into the posterior chamber. Immediately after the delivery, the injector was removed and the incision sutured with a nylon 10-0 C3 suture, ensuring a tight seal and a well-controlled ocular pressure. Retinal tack forceps (Geuder AG) were used to move the prosthesis into place and to advance two custom-made stainless-steel retinal tacks through the prosthesis into the sclera. Then, iris retractors were removed, and all incisions were sutured with resorbable sutures (Vicryl 7-0 GS-9, Ethicon). Celestan® (4 mg ml$^{-1}$) was injected subconjunctivally, and tobramycin/dexamethasone (Tobradex; Alcon) was applied to the eye.

**Statistical analysis and graphical representation**. Statistical analysis and graphical representation were performed with Prism (version 9.3.1; GraphPad Software). Power analysis was performed with G*Power (version 3.1.9.5). The normality test (D'Agostino & Pearson omnibus normality test) was performed in each dataset to justify using a parametric or non-parametric test.

**Reporting summary**. Further information on research design is available in the Nature Research Reporting Summary linked to this article.

## Data availability

The authors declare that the data supporting the findings of this study are available in the paper and its supplementary information files. Source data are provided with this paper.

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

## Acknowledgements

We want to acknowledge the Center of Micronanotechnology (École Polytechnique Fédérale de Lausanne), the Neural Microsystems Platform (Campus Biotech Geneva) and the Preclinical Neuroscience Platform (Campus Biotech Geneva) for their support. We also want to acknowledge Prof. Graham Knott and the Bioelectron Microscopy Core Facility (École Polytechnique Fédérale de Lausanne) for their help with the semithin sections. We wish to thank Prof. Maureen McCall (University of Louisville) for her advice with the IAA model and Anisoara Nicastro (Hôpital Ophtalmique Jules-Gonin) for her help during surgeries. We also wanted to thank Prof. Walid Habre (University of Geneva) and his team (Sylvie Roulet, Jean-Pierre Giliberto, John Diaper and Xavier Belin) for their help with the anaesthesia. This work was supported by École Polytechnique Fédérale de Lausanne (to D.G.), Medtronic plc (to D.G.), Velux Stiftung (Project 1102 to D.G.), Fondation Pro Visu (to D.G.) and Gebert Rüf Stiftung (Project GRS-035/17 to D.G.).

## Author contributions

P.V. performed IAA injection, electrophysiology and histology, and she wrote the manuscript. M.J.I.A.L. designed, fabricated and characterised the POLYRETINA device. C.-H.V. performed OCT imaging and developed the injector. E.G.Z. performed OCT imaging, electrophysiology and histology, and she performed animal handling and healthcare. G.S. fabricated the POLYRETINA device and retinal tacks. T.J.W. performed surgical injections of POLYRETINA. D.G. designed and led the study and wrote the manuscript. All the authors read and accepted the manuscript.

## Competing interests

D.G. and M.J.I.A.L. are authors of a patent covering the POLYRETINA device [Applicants: École Polytechnique Fédérale de Lausanne; Inventors: Ghezzi Diego, Airaghi Leccardi Marta Jole Ildelfonsa and Ferlauto Laura; Application number: WO2018177547A1]. D.G., C.-H.V. and T.J.W. are authors of a patent covering the injector [Applicants: École Polytechnique Fédérale de Lausanne and Fondation Asile des Aveugles; Inventors: Ghezzi Diego, Wolfensberger Thomas and Vila Charles-Henri; Application number: WO2020229683A1]. The remaining authors declare no competing interests.
