## [Peer Review File · Nature Communications]

Reviewer Comments, first round -

Reviewers' comments:

Reviewer #1 (Remarks to the Author):

Vagni et al. POLYRETINA prosthesis restores light responses in-vivo in blind Göttingen minipigs.

The paper develops and characterizes a model of blindness in minipigs on which they test the POLYRETINA implant and measure the electrophysiological responses to photovoltaic stimulation.

There are 4 main parts in this report: 1) introduction and characterization of the blindness model, 2) design of the implant, 3) surgical approach and 4) functional outcome. Overall, parts 1-3) are well presented and data quality is high with novelty involved. However, The functional outcome of stimulation is poorly documented and does not yet support the claim of "restoration of light responses".

Here are some specific comments:

1) Description of the model of blindness: this section is very well documented and data is convincing. The loss of ERG responses is correlated with histological findings at different locations of the retina.

2) POLYRETINA fabrication and testing. This section is well done but would benefit from the photovoltage voltage measurements similarly to what the group has published for their previous version of the device: <https://www.nature.com/articles/s41467-018-03386-7> Figure 4b . This is important to understand the ERG responses in vivo presented in figure 4c of the present paper.

3) The surgical approach is well documented although supplying a video of the procedure would be a tremendous plus.

4) The functional outcomes are disappointing and contrast with the rest of the paper. They are supposed to support the claim of functional restoration of light responses.

- Please clarify the number of animals and the distribution on the different groups: control, IAA for histology and IAA for implantation. Why were only 2 eyes implanted with POLYRETINA? Are the 2 eyes from one rabbit only? Were there other implantations that failed?

- Figure 4e: why is the Before IAA condition so different from the control VEPs in figure 1k?

- The removal of the recording wires between recording sessions is a significant confounding factor. I would suggest repeating the experiment with implanted screw. The skin can be stitched and opened back above the screw to avoid any infection.

- Figure 1g: could you please plot the raw traces in response to the 3 amplitudes as well as the negative peak that was used to compute the fEEP amplitude?

- Can you explain why the latency in cortical responses is longer with photovoltaic stimulation than with visual responses?

- Eye 1 gives higher responses than pre implantation. Eye 2 gives an amplitude reduced by x3 compared to healthy timepoint. This does not support the claim of restoration of light responses.

5) No discussion is provided. Although my personal opinion is in favor of short discussions. It is useful to place the findings into context, mention the limitations (spot size and visual acuity for example), biocompatibility etc...

In conclusion, the functional outcome data is not convincing and would require:

- 4-5 implanted eyes total. Please provide a statistical plan before repeating the experiment.
- cranial electrodes that are fixed to the skull and not replaced at every session.
- improved presentation of the raw traces in response to stimulation. Emphasize the points used to quantify the amplitude.

The rest of the paper is of high quality and demonstrates the relevance of the model and the success of the surgical approach.

Reviewer #2 (Remarks to the Author):

The authors examine the effect of transplanting an epiretinal device in pigs treated with IAA to damage photoreceptors.

IAA modifies GAPDH leading to a partial inhibition of glycolysis, based on dosage. This effect is transient. While most organs depend upon glycolysis, the high metabolic rate of photoreceptors makes them particularly susceptible to its inhibition. This transient inhibition of glycolysis is toxic to rods, and the authors show loss of RHO-expressing rods. Initially, cones lose outer segments and ERG function, but they tend to recover and do not die. And, the surviving cones regain ERG function. The histologic analysis would benefit from higher power views, co-staining with outer segment and inner segment markers and EM to evaluate the structures. The authors refer repeatedly to the pigs as blind following IAA treatment. But, their own histology and seemingly recovery of photopic ERG might suggest otherwise, even though neither are a measure of vision. This might be an important point given their reliance on cone wavelength stimulation to evaluate the implant. The authors should provide evidence that the pigs are blind following IAA treatment. This does not have to be a detailed analysis, but could include videos of the pigs following a laser pointer on a wall or perhaps moving through obstacles such as cones placed between the animals and their food in a lighted room.

The basic premise seems to be found in Lines 52-56: to assess the functionality of device, deprive retina of photoreceptor input, this ensures signal comes from device...

However, Line 107-113: although rod loss, cone preservation up to 4m with only outer segment loss.

The ERG and L/M opsin data seems to demonstrate recovery of cones following IAA. Here lies the confound. If cone function returns (ERG recovery at 2 months in panel b of Figure 4), especially in regions 2mm from the optic nerve (Supp. Figures 1 & 2), then how do they evaluate a signal from the device after it is implanted 3.5 months after IAA-treatment? After showing a partially recovered cone signal 2 months after IAA in Figure 4b, Figure 4c shows corneal potentials evoked using yellowish-green (565nm) light at various flash durations, supposedly demonstrating signals solely acquired from the device 2 weeks post-implantation. Lines 213-217 attempt to explain the reasoning for this but I don't understand the rationale.

Figure 4e-g is the crux of the paper. There are several potential problems here. Without control data showing the implanted eye as 'rebounded' and the control eye as 'diminished', Figure 4e-g looks like potential post-IAA cone recovery after 2 months. Also, at 2 months after IAA the VEP is flat, immediately after implantation and 2 weeks post-implant the VEP had recovered. However, no pre-surgical baseline data was presented, so a VEP signal could have been there before the device was implanted. Too few animals evaluated. The stimulus is unfortunately focused on mimicking a cone response and not mimicking a rod response. This might be impactful because rods appear to be lost in the model, but cones are not.

Reviewer #3 (Remarks to the Author):

The authors present work on an advanced retinal implant which they call POLYRETINA. This retinal implant shall provide a larger area of stimulation (i.e. visual field for patients) because it is large and flexible as it is placed within the vitreous cavity on the retina (epiretinally).

The authors present a truly impressive pile of data in order to prove their point of feasibility as a (last?) step before going to human trials. They present the device, including the manufacturing technique. They present the animal model, including how to make it blind, the histological proof of the latter, the electrophysiological data. They present the surgical technique for insertion. They present functional data, in vivo assessments, including rather advanced VEP recordings in pigs. Each of these astonishing accomplishments is worth a publication of its own.

The authors belong to a highly regarded institution with a longstanding history in retinal implants. The work is new and important. If all holds true it could constitute a significant step for patients suffering from retinal degenerative disease.

However – I had much difficulty reading the manuscript. In many areas the language is deeply colloquial. And long. In other areas, I feel important data for understanding the work was missing. In my opinion, the authors try to unnecessarily collect all the above mentioned work into ONE manuscript. It feels much too crowded to be able to pay attention to the pile of data collected. I feel it could be much better to divide the manuscript into four: 1. The new device, 2. The animal model, 3. The surgical technique, 4. The results. It seems impossible to me (after reading the current version) to clearly describe all things in one manuscript and give the deserved detail to each task and accomplishment in this very condensed form (although, as mentioned above, I feel it mostly lengthy and NOT condensed.).

I could not follow the content, its importance, the line of thought all along, although I am from this very field of research.

I have a very hard time believing that the device is in close contact with the retina in all areas, with all electrodes. And if so, why should there be no damage to the retina due to pressure of the device onto it?

In its current form, I am sorry to say that, I cannot recommend it for publication. It should be completely rewritten, made more concise, give more detail where adequate, make us of less fill-in-phrases and possibly be separated into several manuscripts.

Also, in comparison to other work in the field the ranking of the chosen journal seems very high.

Reviewer #1 (Remarks to the Author):

Vagni et al. POLYRETINA prosthesis restores light responses in-vivo in blind Göttingen minipigs.

The paper develops and characterizes a model of blindness in minipigs on which they test the POLYRETINA implant and measure the electrophysiological responses to photovoltaic stimulation.

Re: we thank the reviewer for the valuable comments.

There are 4 main parts in this report: 1) introduction and characterization of the blindness model, 2) design of the implant, 3) surgical approach and 4) functional outcome. Overall, parts 1-3) are well presented and data quality is high with novelty involved. However, The functional outcome of stimulation is poorly documented and does not yet support the claim of "restoration of light responses".

Here are some specific comments:

1) Description of the model of blindness: this section is very well documented and data is convincing. The loss of ERG responses is correlated with histological findings at different locations of the retina.

Re: we thank the reviewer for the positive assessment of these experiments.

2) POLYRETINA fabrication and testing. This section is well done but would benefit from the photovoltage measurements similarly to what the group has published for their previous version of the device: <https://www.nature.com/articles/s41467-018-03386-7> Figure 4b . This is important to understand the ERG responses in vivo presented in figure 4c of the present paper.

Re: detailed characterisation of photovoltage measurements with Ti/TiN pixels have been recently published in another paper¹ (Ref 17 in the revised manuscript). In our opinion, it is not useful to repeat those measures here. Also, the corneal recordings of the potential generated by the prosthesis activation have been removed since they were not necessary for the scope of the paper.

3) The surgical approach is well documented although supplying a video of the procedure would be a tremendous plus.

Re: we included a short video of the surgery (see Supplementary Video 1).

4) The functional outcomes are disappointing and contrast with the rest of the paper. They are supposed to support the claim of functional restoration of light responses.

Re: please find below our answers/changes.

- Please clarify the number of animals and the distribution on the different groups: control, IAA for histology and IAA for implantation. Why were only 2 eyes implanted with POLYRETINA? Are the 2 eyes from one rabbit only? Were there other implantations that failed?

Re: for each experiment, we clarified the number of animals/eyes used. There were no implantations that failed. Simply, there was a standard surgical learning curve. We needed some animals to practice and optimise the surgery, the injector and the retinal tacks. We still included these animals in the first characterisation (IAA-treatment) but not in the recovery study with POLYRETINA.

Please note that we removed the original two eyes from the revised manuscript because of the issue mentioned below by the reviewer (cortical electrode removal and replacement).

- Figure 4e: why is the Before IAA condition so different from the control VEPs in figure 1k?

Re: illuminations are different in the two experiments. We used a flash white Ganzfeld stimulator close to the eye for ERG/VEP characterisation before/after IAA (Figs. 2 and 3 in the revised manuscript). We used a green LED focalised on the cornea for VEP and EEP before/after POLYRETINA (Fig. 7 in the revised manuscript). Illumination efficiency is higher for the Ganzfeld stimulator.

Also, differences might be attributed to animal variability and, as mentioned by the reviewer, electrode removal and replacement (proximity to the cortex and exact location).

- The removal of the recording wires between recording sessions is a significant confounding factor. I would suggest repeating the experiment with implanted screw. The skin can be stitched and opened back above the screw to avoid any infection.

Re: over the years, we used many types of electrodes, including implanted screws into the skull. The problem was not an infection, which could be prevented. The problem was that even if implanted in a few cm of bone, minipigs kept scratching, reopening the wound, until they would have removed electrodes/screws, and most of the time did that. Please note that suturing multiple times the minipig skin is not a trivial step. Also, opening and closing the wound multiple times is not ideal because the healing process is impaired. Then, we used implanted k-wires which can be inserted through the skin without openings, but they also removed the wires. When animals remove the screw/wire, they damage the bone, making a second implant a challenge. This point is the reason why we decided to remove the wires after each surgery. We understand that this choice might be an uncontrolled variable in the data. Therefore, in the revised manuscript, we decided to test recovered light responses with POLYRETINA only acutely, before, during and after surgery with k-wires left in place (no removal) between recordings.

- Figure 1g: could you please plot the raw traces in response to the 3 amplitudes as well as the negative peak that was used to compute the fEEP amplitude?

Re: we displayed the raw traces in Fig. 7. fVEP and fEEP were computed as peak-to-peak amplitudes (method indicated in Fig. 3).

- Can you explain why the latency in cortical responses is longer with photovoltaic stimulation than with visual responses?

Re: in our dataset, responses to natural stimulation and photovoltaic stimulation have comparable latency. This result is coherent with our previous findings. In the field of retinal prostheses, there is the notion that cortical potential elicited by prosthetic stimulation have a shorter latency of about 10 to 20 ms than natural responses since phototransduction is avoided². However, this argument is not always true. When the epiretinal stimulation mechanism only targets axonal fibres (direct stimulation), the latency is shorter than natural stimulation. Also, it is the case for subretinal stimulation that primarily targets bipolar cells, but not for epiretinal network-mediated stimulation (like POLYRETINA). We previously demonstrated that POLYRETINA activates the retina via a network mediated process targeting bipolar and amacrine cell³. Our data with explanted blind retinas^{1,3,4} showed that spikes in retinal ganglion cells elicited by POLYRETINA have longer latency (by approx 10-20 ms) than comparable recordings with subretinal photovoltaic stimulation⁵. This evidence is why we expect responses to natural stimulation and POLYRETINA stimulation to have comparable latency, as in our dataset. We believe that the explanation for this behaviour is linked to the strong activation of amacrine cells during epiretinal network mediated stimulation which provides inhibition to retinal ganglion cells. This hypothesis was verified with retinal modelling in our previous paper (see figure 12a,b in³). Network mediated epiretinal stimulation induces responses in retinal ganglion cells with longer latency than subretinal stimulation (see figure 12b in³), possibly because the excitation/inhibition ratio is different between subretinal and epiretinal network mediated stimulation. Subretinal stimulation induced a balanced contribution of excitation (from bipolar cells) and inhibition (from amacrine cells), while epiretinal stimulation induces a stronger contribution from amacrine cells than bipolar cells. Hence the response delay is higher for epiretinal network mediated stimulation.

- Eye 1 gives higher responses than pre implantation. Eye 2 gives an amplitude reduced by x3 compared to healthy timepoint. This does not support the claim of restoration of light responses.

Re: restoration of light sensitivity means that light responses with the device are higher than the implanted blind condition, which is the case for both eyes. However, as already pointed out, the reposition of the electrode also might be a confusing factor. Therefore, these data were removed and replaced with a pre/post surgery dataset obtained without removing electrodes.

5) No discussion is provided. Although my personal opinion is in favor of short discussions. It is useful to place the findings into context, mention the limitations (spot size and visual acuity for example), biocompatibility etc...

Re: we provided a balanced discussion of our results, highlighting advancements but also limitations.

In conclusion, the functional outcome data is not convincing and would require:

- 4-5 implanted eyes total. Please provide a statistical plan before repeating the experiment.

Re: we repeated the experiments in 3 additional animals (1 eye per animal), but we decided to remove the original two for the reason mentioned above by the reviewer. Statistical analysis (Fig. 7c in the revised manner) was performed and confirmed the sufficient number of animals. Power analysis has been performed to determine the sample size of 3 (based on a required power higher than 0.90 and alpha error probability 0.05). In the end, the actual test power resulted in 0.9987.

- cranial electrodes that are fixed to the skull and not replaced at every session.

Re: unfortunately, this is not technically feasible in our hands with large animals like minipigs. To avoid this possible confusing factor in POLYRETINA evaluation, the old data were removed and replaced with a pre/post surgery dataset obtained without removing electrodes.

- improved presentation of the raw traces in response to stimulation. Emphasize the points used to quantify the amplitude.

Re: traces are shown, and points for quantification are shown.

The rest of the paper is of high quality and demonstrates the relevance of the model and the success of the surgical approach.

Re: we thank the reviewer for the assessment.

CITED PAPERS

1. Chenais, N. A. L., Airaghi Leccardi, M. J. I. & Ghezzi, D. Photovoltaic retinal prosthesis restores high-resolution responses to single-pixel stimulation in blind retinas. *Commun Mater* 2, 28 (2021).
2. Mandel, Y. *et al.* Cortical responses elicited by photovoltaic subretinal prostheses exhibit similarities to visually evoked potentials. *Nature communications* 4, 1980 (2013).
3. Chenais, N. A. L., Leccardi, M. J. I. A. & Ghezzi, D. Capacitive-like photovoltaic epiretinal stimulation enhances and narrows the network-mediated activity of retinal ganglion cells by recruiting the lateral inhibitory network. *J. Neural Eng.* 16, 066009 (2019).
4. Ferlauto, L. *et al.* Design and validation of a foldable and photovoltaic wide-field epiretinal prosthesis. *Nature Communications* 9, 992 (2018).
5. Lorach, H. *et al.* Photovoltaic restoration of sight with high visual acuity. *Nat Med* 21, 476–482 (2015).

Reviewer #2 (Remarks to the Author):

The authors examine the effect of transplanting an epiretinal device in pigs treated with IAA to damage photoreceptors.

Re: we thank the reviewer for the valuable comments.

IAA modifies GAPDH leading to a partial inhibition of glycolysis, based on dosage. This effect is transient. While most organs depend upon glycolysis, the high metabolic rate of photoreceptors makes them particularly susceptible to its inhibition. This transient inhibition of glycolysis is toxic to rods, and the authors show loss of RHO-expressing rods. Initially, cones lose outer segments and ERG function, but they tend to recover and do not die. And, the surviving cones regain ERG function.

Re: in a previous paper with IAA in pigs (Ref 39 in the manuscript, lines 120-121), ERG responses recovered over time (5-6 weeks) after IAA administration. To rule out this possibility, in the revised manuscript, we quantified the a- and b- waves over time (Fig. 2g), similar to what we did with OCT (Supp. Fig. 2). In the tested animals/eyes, we did not observe any recovery of light-adapted cone-driven ERG until up to 2 to 3 months (80 days) after IAA administration. Please note that we made additional experiments/changes in the revised manuscript to substantiate our claim of recovery of light sensitivity with POLYRETINA (see below).

The histologic analysis would benefit from higher power views, co-staining with outer segment and inner segment markers and EM to evaluate the structures.

Re: in the revised manuscript, we provided additional stainings to complement our analysis. In particular, we included s-opsin and Na⁺/K⁺ ATPase stainings. We attempted performing sections for cryoEM, but the experiment failed. Therefore, we included optical images of semithin sections, which shows retinal cells in much greater detail (Fig. 1f). Most importantly, we performed the histology analysis for the three minipigs/eyes used in the functional recovery experiments with POLYRETINA (Fig. 9). Those three animals showed no residual marker for none of the tested stainings indicating complete blindness.

The authors refer repeatedly to the pigs as blind following IAA treatment. But, their own histology and seemingly recovery of photopic ERG might suggest otherwise, even though neither are a measure of vision. This might be an important point given their reliance on cone wavelength stimulation to evaluate the implant.

Re: we would like to clarify that our data never showed ERG recovery. What was shown in the original manuscript (now removed) was the potential generated by the prosthesis recorded from the corneal electrode, not a remaining/recovered ERG. Indeed, the shape of the signal was not the one of an ERG but corresponding to the capacitive voltage generated by the device (this type of signal was reported in other papers related to retinal implants). We removed this data since it is not helpful towards the goal of the study.

There is another point to be clarified (lines 85-86 in the revised manuscript) concerning our histological analysis. The image sequences (in Supp. Fig. 4-17) looks like a longitudinal study, but of course, each time point is from a different animal/eye since it is a post mortem analysis. Even though it might be tempting to say that there is a recovery of cones at the 2.5M time point, in reality, what those images show is likely an inter-subject variability of IAA (explained at lines 99-100 in the revised manuscript). The animal at the 2.5 months time point had stronger resistance to IAA than the others, as visible from the preservation of s- and l/m opsins. But, such preservation is not present in the animal at a later time point (3 months), confirming the hypothesis of an inter-subject variability. Therefore, to mitigate this variability, the animals tested with POLYRETINA were injected twice. Histological images on the minipigs implanted with POLYRETINA (Fig. 9) showed no residual marker for none of the tested stainings indicating complete blindness.

The authors should provide evidence that the pigs are blind following IAA treatment. This does not have to be a detailed analysis, but could include videos of the pigs following a laser pointer on a wall or perhaps moving through obstacles such as cones placed between the animals and their food in a lighted room.

Re: in the revised manuscript, we provided several pieces of evidence that animals are light insensitive. Both histological and electrophysiological experiments support this claim. Most importantly, in Fig. 7, response to light was compared immediately before and immediately after the placement of POLYRETINA. We can now exclude the hypothesis of a response recovery due to the recovery of cones since the experiments were conducted the same day.

Regarding behavioural testing, this is not a relevant experiment in our opinion since it would not be possible to perform the same experiment with POLYRETINA for comparison. Nevertheless, we performed a simplified behavioural task during which animals should find food in the area outside the cage. Overall, blind animals took much longer than control animals to find food (few seconds for controls and tens of seconds for blind). We repeated this experiment several times so we have data, but we are not in favour of adding them to the manuscript since this experiment is not directly linked to the validation of POLYRETINA, which was done by electrophysiological means. However, we could add them if the reviewer thinks they are necessary.

The basic premise seems to be found in Lines 52-56: to assess the functionality of device, deprive retina of photoreceptor input, this ensures signal comes from device...However, Line 107-113: although rod loss, cone preservation up to 4m with only outer segment loss. The ERG and L/M opsin data seems to demonstrate recovery of cones following IAA. Here lies the confound. If cone function returns (ERG recovery at 2 months in panel b of Figure 4), especially in regions 2mm from the optic nerve (Supp. Figures 1 & 2), then how do they evaluate a signal from the device after it is implanted 3.5 months after IAA-treatment? After showing a partially recovered cone signal 2 months after IAA in Figure 4b, Figure 4c shows corneal potentials evoked using yellowish-green (565nm) light at various flash durations, supposedly demonstrating signals solely acquired from the device 2 weeks post-implantation. Lines 213-217 attempt to explain the reasoning for this but I don't understand the rationale.

Figure 4e-g is the crux of the paper. There are several potential problems here. Without control data showing the implanted eye as 'rebounded' and the control eye as 'diminished', Figure 4e-g looks like potential post-IAA cone recovery after 2 months. Also, at 2 months after IAA the VEP is flat, immediately after implantation and 2 weeks post-implant the VEP had recovered. However, no pre-surgical baseline data was presented, so a VEP signal could have been there before the device was implanted. Too few animals evaluated. The stimulus is unfortunately focused on mimicking a cone response and not mimicking a rod response. This might be impactful because rods appear to be lost in the model, but cones are not.

Re: we repeated the functional experiments with POLYRETINA making several changes. Most importantly, as suggested by the reviewer, pre/post comparison is made the same day, immediately before and after placement of the device. Last, histological analysis of the implanted animals showed that both rods and cones have degenerated.

In summary, to substantiate our claim of restoration of light sensitivity due to POLYRETINA implantation, we have performed the following experiments:

1. Longitudinal ERGs (Fig. 2g) showed that there is no ERG recovery in a period of 2-3 months.
2. We demonstrated the functional recovery with POLYRETINA the same day with a pre/post analysis (Fig. 8). In this manner, the recovered responses cannot be attributed to cone recovery or survival, but only to the presence of POLYRETINA.
3. We included a more detailed histological analysis of the treated animals (Fig. 1 and related Supplementary Figures). We specified that the IAA treatment might have some inter-animal variability. Therefore, in the animals tested with POLYRETINA, we repeated the IAA administration twice to ensure complete degeneration of rods and cones. The post-mortem analysis on the animals tested with POLYRETINA showed no sign of any tested marker for both cones and rods (Fig. 9), indicating that they were fully blind.

These results substantiate our claims and rule out the hypothesis of response recovery due to cone recovery or survival.

Reviewer #3 (Remarks to the Author):

The authors present work on an advanced retinal implant which they call POLYRETINA. This retinal implant shall provide a larger area of stimulation (i.e. visual field for patients) because it is large and flexible as it is placed within the vitreous cavity on the retina (epiretinally).

The authors present a truly impressive pile of data in order to prove their point of feasibility as a (last?) step before going to human trials. They present the device, including the manufacturing technique. They present the animal model, including how to make it blind, the histological proof of the latter, the electrophysiological data. They present the surgical technique for insertion. They present functional data, in vivo assessments, including rather advanced VEP recordings in pigs. Each of these astonishing accomplishments is worth a publication of its own.

Re: we thank the reviewer for the assessment of the paper and the valuable comments.

The authors belong to a highly regarded institution with a longstanding history in retinal implants. The work is new and important. If all holds true it could constitute a significant step for patients suffering from retinal degenerative disease.

However – I had much difficulty reading the manuscript. In many areas the language is deeply colloquial. And long. In other areas, I feel important data for understanding the work was missing. In my opinion, the authors try to unnecessarily collect all the above mentioned work into ONE manuscript. It feels much too crowded to be able to pay attention to the pile of data collected. I feel it could be much better to divide the manuscript into four: 1. The new device, 2. The animal model, 3. The surgical technique, 4. The results. It seems impossible to me (after reading the current version) to clearly describe all things in one manuscript and give the deserved detail to each task and accomplishment in this very condensed form (although, as mentioned above, I feel it mostly lengthy and NOT condensed.).

I could not follow the content, its importance, the line of thought all along, although I am from this very field of research.

Re: we rewrote most of the manuscript to make it more readable. We paid attention to your handwritten comments. However, we agreed with the Editor to not split the data into several manuscripts.

I have a very hard time believing that the device is in close contact with the retina in all areas, with all electrodes. And if so, why should there be no damage to the retina due to pressure of the device onto it?

Re: we performed two experiments to check for the close contact of the device. Unfortunately, after the placement of the device, OCT is not possible anymore. Therefore we took several echographies of the implanted eyes (representative example in Fig. 6I). Next (following a request from the editor), we wanted to validate that different points of the device over its surface could activate the retina independently. The only way we could do that was by performing focused intraocular stimulation. Multifocal visual stimulation was not possible since we removed the lens during surgical implantation. Therefore, we used an intraocular optic fibre to stimulate different points of the device (Fig. 8). Our results showed that all the randomly selected points of the device could activate the retina and generate a detectable EEP.

In its current form, I am sorry to say that, I cannot recommend it for publication. It should be completely rewritten, made more concise, give more detail where adequate, make us of less fill-in-phrases and possibly be separated into several manuscripts.

Also, in comparison to other work in the field the ranking of the chosen journal seems very high.

Re: the manuscript has been rewritten.

Reviewer Comments, second round -

Reviewer #1 (Remarks to the Author):

The authors made a great effort to respond to the many concerns raised in the previous version and completely renewed their electrophysiology dataset.

The characterization of the retinal degeneration model and the device fabrication / implantation sections are well presented and the surgery video is very informative.

The crucial point raised by 2 of the 3 reviewers was to know whether cortical responses to electrical stimulation are actually evoked by the implant or by remaining photoreceptors.

Based on 3 new animals by comparing preimplantation and post implantation VEP, it seems that the cortical response increased at a given irradiance level (BTW figure says $890\mu\text{W}\cdot\text{mm}^{-2}$ but the text says 1.18 mW mm^{-2} , which one is it?,).

A few questions remain to be convinced of this conclusion:

- Was the pre-implantation stimulation performed just before insertion of the implant or before the vitrectomy and lens removal?
- You do have the amplitude/irradiance curve at many irradiances (fig 7b), why not show the 3 curves of each animals rather than the bar plot at an arbitrary irradiance value that could have been chosen a posteriori to show the strongest difference.
- Can you please plot figure 7a over 2 consecutive stimulation periods (meaning only 100 repetitions of 2s) ?

If those questions can be addressed, the claim of "restoration" of light response can be made and I would support the publication.

Reviewer #2 (Remarks to the Author):

The authors have responded to criticism by throwing out previous data and adding new eyes. They have also investigated their IAA model of vision loss more completely using both photopic and scotopic ERGs at different light levels. It is disappointing that they did not evaluate the experimental eyes in the same fashion. They only used a single wavelength of light. 565 nm. The number of experimental eyes is still quite limited to reach the kind of conclusions stated.

Reviewer #3 (Remarks to the Author):

The MS has much improved since I last read it. Readability and preciseness have improved.

I have almost no minor points such as grammar or spelling - these will be checked before print by the publisher.

My concerns are that the MS still seems crowded to me, too much stuffed into it, i.e. establishing an animal model, improving an implant, improving the surgery for implantation, measuring function, establishing SD-OCTs in these animals, establishing EEP-measurement. - All of these accomplishments would have merited a single publication of its own.

This crowded presentation might be the reasons for some awkward 'jumps', at times the MS is extremely detailed, at other times it jumps without me being able to follow. - Isn't the reason for a publication to let readers be able to re-enact the research and come to the same conclusions. Impossible here, even if one had the implant.

I would like to see an image of an implant in an eye after implantation, where I can assess how the implant fits the curve of the eyeball. This is my absolutely major concern regarding the

feasibility of this approach: does the implant really touch the retina everywhere? Everywhere with the same distance and pressure? What will this pressure do to the retina in the long term?

I have great difficulty assessing the number of animals in each treatment arm. It is not absolutely clearly stated how many animals served where. Was it really only one animal in the establishing degeneration trial? Was there any power analyses performed? Why not give a table where the reader can follow what was performed in which animal with which results? There seems to be a lack in rigorous analysis. Or in will of publishing it. The task is great now doubt, but it would easier to judge it even higher when we get the feeling that the authors have set very high standards for themselves by publishing all the data (not merely examples), by self-critically examining the results. No need to fear, the accomplishments are great!

There is a lack of discussing critically the results. Strengths? Weaknesses (honestly, what lacks?, what misses? What failed? What will be the difficulties in the future? In humans? Costs? Number of animals?). I get the impression that the authors fear a rejection if this all would be critically illuminated. But that is (at least for me) not the case. I want the truths, and the harder the authors are with themselves, the more I trust them. Instead the authors lose themselves in bloomy descriptions and difficult sentences.

Replies to the comments are in blue. Amendments to the manuscript are in red.

Reviewer #1 (Remarks to the Author):

The authors made a great effort to respond to the many concerns raised in the previous version and completely renewed their electrophysiology dataset. The characterization of the retinal degeneration model and the device fabrication / implantation sections are well presented and the surgery video is very informative.

Re: we thank the reviewer for the positive assessment.

The crucial point raised by 2 of the 3 reviewers was to know whether cortical responses to electrical stimulation are actually evoked by the implant or by remaining photoreceptors.

Based on 3 new animals by comparing preimplantation and post implantation VEP, it seems that the cortical response increased at a given irradiance level (BTW figure says 890uW.mm-2 but the text says 1.18 mW mm-2, which one is it?,).

Re: yes this is a mistake, the correct number is 1.18 mW mm-2. We made the necessary change in the caption of the figure.

A few questions remain to be convinced of this conclusion:

- Was the pre-implantation stimulation performed just before insertion of the implant or before the vitrectomy and lens removal?

Re: pre-implantation test was made before starting the surgery. It means before vitrectomy and lensectomy. We clarified this point in the text (Page 5, "Recordings before surgery (i.e. before lensectomy and vitrectomy)...". In our opinion, this is not a major difference since the transparency of the lens/vitreous is higher than 95% in the green region (565 nm) of the spectrum [1]. So removal of the lens and vitreous would not affect the overall transmittance of the eye media. Conversely, removing the lens might reduce the light collection capability (our LED is collimated and not focused). In short, data post-implantation might be penalised by the absence of the lens (less light collected to the retina/POLYRETINA compared to pre-surgery).

- You do have the amplitude/irradiance curve at many irradiances (fig 7b), why not show the 3 curves of each animals rather than the bar plot at an arbitrary irradiance value that could have been chosen a posteriori to show the strongest difference.

Re: we have replaced previous Fig. 7b with Fig. 7f which shows data averaged across the 3 animals. Statistical analysis has been therefore updated [Page 6, "After POLYRETINA implantation, fEEPs could be measured (Fig. 7d, red traces; averages of 200 consecutive responses) with peak-to-peak amplitudes (Fig. 7f, red circles) above the level of the peak-to-peak biological fluctuation (Fig. 7f, grey filled area between dashed lines). Peak-to-peak amplitudes in fEEPs follow a semi-log trend as a function of the irradiance. The data before and after POLYRETINA implantation are represented by two statistically different curves ($p < 0.0001$; extra sum-of-squares F test)."].

We would like to clarify that 1 mW mm⁻² is not an 'arbitrary irradiance value' chosen to see a stronger effect. 1 mW mm⁻² is the working irradiance set for POLYRETINA since our first paper in 2018 [2].

The reasoning behind this choice was spelled out in the text (now at page 6). In a photovoltaic retinal prosthesis (under real operation), response modulation using amplitude modulation (of the irradiance) is not trivial. A photovoltaic retinal prosthesis requires AR/projecting glasses (see PRIMA from pixium vision). In those glasses the laser output is a fixed parameter. If needed, modulation can be achieved by modulating the pulse duration controlling the time of the DMD mirrors in the projector (but this is another story).

We choose 1 mW mm⁻² (here 1.18) as working irradiance for two reasons:

(1) it is below the safety limit, which is approx 2 mW mm⁻² [1].

(2) RGC response measured in retinal explants saturates around this value [2,3].

Because of the reasons above, the quantification really makes sense at this value. We further clarify these aspects in the manuscript [Page 6, "Our previously published results^{11,17} identified 1 mW mm⁻² as the irradiance level within safety limits leading to a saturated response in mouse RGCs upon POLYRETINA activation. Therefore, we selected this value as standard irradiance for the functional activation of POLYRETINA. Here, we statistically compared the recovery of light sensitivity at the closest irradiance that we could obtain (1.18 mW mm⁻²). The statistical analysis revealed that peak-to-peak amplitudes are significantly higher after implantation compared to before implantation ($p = 0.0074$, power = 1.00; two-tailed paired t-test). Also, the statistical analysis across all the irradiance levels showed that the first irradiance level leading to a statistically significant higher peak-to-peak amplitude is 33 $\mu\text{W mm}^{-2}$ ($p = 0.0308$, power = 1.00; two-tailed paired t-test). This value is coherent with the activation threshold we previously measured for full-field stimulation in retinal explants from blind mice (47.35 $\mu\text{W mm}^{-2}$)¹¹."].

- Can you please plot figure 7a over 2 consecutive stimulation periods (meaning only 100 repetitions of 2s) ?

Re: yes. data is plotted in the new **Fig. 7e**. Averaged fEEPs are qualitatively similar among the two blocks.

If those questions can be addressed, the claim of "restoration" of light response can be made and I would support the publication.

Re: we believed to have answered the reviewer's questions.

Cited papers

[1] EDWARD A. BOETTNER, J. REIMER WOLTER; Transmission of the Ocular Media. Invest. Ophthalmol. Vis. Sci. 1962;1(6):776-783.

[2] Ferlauto, L., Airaghi Leccardi, M.J.I., Chenais, N.A.L. et al. Design and validation of a foldable and photovoltaic wide-field epiretinal prosthesis. Nat Commun 9, 992 (2018). <https://doi.org/10.1038/s41467-018-03386-7>

[3] Chenais, N.A.L., Airaghi Leccardi, M.J.I. & Ghezzi, D. Photovoltaic retinal prosthesis restores high-resolution responses to single-pixel stimulation in blind retinas. Commun Mater 2, 28 (2021). <https://doi.org/10.1038/s43246-021-00133-2>

Reviewer #2 (Remarks to the Author):

The authors have responded to criticism by throwing out previous data and adding new eyes. They have also investigated their IAA model of vision loss more completely using both photopic and scotopic ERGs at different light levels.

Re: we believed reviewers have expressed fair and constructive criticisms, so we modified our dataset accordingly.

It is disappointing that they did not evaluate the experimental eyes in the same fashion.

Re: we respectfully disagree with the reviewer. Experimental animals were evaluated in the same way as the batch used for the degeneration study. Both batches had IHC analysis to verify degeneration of photoreceptors (**Fig. 1** and **Fig. 9** respectively). The key difference was in the data shown for electrophysiological evaluation. For the experimental minipigs, we wanted to focus the attention in the manuscript on the crucial point for our study, which is the evaluation of POLYRETINA (see next point). Regardless, the animals were still subjected to the rigorous electrophysiological evaluation set to check blindness before surgical implantation, but we did not include those data. In response to the reviewer's comment, in the modified manuscript, we now included the results of dark-adapted and light-adapted fERGs and fVEPs with Ganzfeld stimulator, before and after IAA administration (**Fig. 7a-c** and text at page 5).

We would like to clarify that dark-adapted and light-adapted fERGs and fVEPs with Ganzfeld stimulator were performed before IAA injection and two weeks before the surgery. It is important to note that this set of experiments requires approximately 3 hr to be performed (including the dark adaptation). Therefore, it was absolutely not possible for us to do it on the same day of the surgery, which is already an intense and long

experimental day with pre-surgical recordings, surgery and post-surgical recordings. The day of the surgery was focused exclusively on the POLYRETINA evaluation. For this reason, it was done two weeks before the surgery. 2-weeks is the minimum time required in our license between two surgeries for animals to recover.

They only used a single wavelength of light. 565 nm.

Re: yes, and this is correct. POLYRETINA is a photovoltaic retinal implant. As such, it works optimally at a specific wavelength (around 560 nm). The light-absorbing layer in the pixel has an absorption spectrum centered around this number, therefore testing at other wavelengths would not be useful because for sure results will be worse. Indeed, in all our papers POLYRETINA is tested only at this wavelength [2,3]. In a clinical evaluation in patients, this is the wavelength (~565 nm) that will be used. Therefore, it is required that preclinical data (needed to move forward to a clinical study) are performed in the same testing condition.

For comparison, other photovoltaic retinal implants follow the same approach. For example, the PRIMA subretinal implant is tested only at one wavelength in the infrared (wavelength 880 nm) [4,5].

The number of experimental eyes is still quite limited to reach the kind of conclusions stated.

Re: the number of experimental animals in a scientific study is determined by statistical considerations. Sample size was determined with a priori power analysis (alpha = 0.05, power 0.85). All tests reported in the study showed that the planned number of animals is sufficient to achieve high statistical results with high power (achieved power of 0.98 or higher). Therefore, we believe that the number of animals is adequate from a statistical point of view, and use of new animals cannot be justified.

Cited papers

[2] Ferlauto, L., Airaghi Leccardi, M.J.I., Chenais, N.A.L. et al. Design and validation of a foldable and photovoltaic wide-field epiretinal prosthesis. *Nat Commun* 9, 992 (2018). <https://doi.org/10.1038/s41467-018-03386-7>

[3] Chenais, N.A.L., Airaghi Leccardi, M.J.I. & Ghezzi, D. Photovoltaic retinal prosthesis restores high-resolution responses to single-pixel stimulation in blind retinas. *Commun Mater* 2, 28 (2021). <https://doi.org/10.1038/s43246-021-00133-2>

[4] Lorach, H., Goetz, G., Smith, R. et al. Photovoltaic restoration of sight with high visual acuity. *Nat Med* 21, 476–482 (2015). <https://doi.org/10.1038/nm.3851>

[5] Prévot, PH., Gehere, K., Arcizet, F. et al. Behavioural responses to a photovoltaic subretinal prosthesis implanted in non-human primates. *Nat Biomed Eng* 4, 172–180 (2020). <https://doi.org/10.1038/s41551-019-0484-2>

Reviewer #3 (Remarks to the Author):

The MS has much improved since I last read it. Readability and preciseness have improved. I have almost no minor points such as grammar or spelling - these will be checked before print by the publisher.

Re: we thank the reviewer for the positive assessment.

My concerns are that the MS still seems crowded to me, too much stuffed into it, i.e. establishing an animal model, improving an implant, improving the surgery for implantation, measuring function, establishing SD-OCTs in these animals, establishing EEP-measurement. - All of these accomplishments would have merited a single publication of its own.

Re: we understand the comment of the reviewer. However, as we wrote in the previous reply, the Editor requested us to not split the data into several manuscripts; part of his private message says "Editorially, we believe that the manuscript is suitable to be presented as one coherent article.." We tried making the manuscript more readable. If the reviewer feels that parts of the text still need improvement, they can point out these sections and we will be happy to make changes.

This crowded presentation might be the reasons for some awkward ‘jumps’, at times the MS is extremely detailed, at other times it jumps without me being able to follow. - Isn't the reason for a publication to let readers be able to re-enact the research and come to the same conclusions. Impossible here, even if one had the implant.

Re: we modified the manuscript to add more details. However we believe that our methodological section is extremely detailed. Other improvements are possible if the reviewer can point out where.

I would like to see an image of an implant in an eye after implantation, where I can assess how the implant fits the curve of the eyeball. This is my absolutely major concern regarding the feasibility of this approach: does the implant really touch the retina everywhere? Everywhere with the same distance and pressure? What will this pressure do to the retina in the long term?

Re: the image of the implant after implantation was already provided in **Fig. 6I**. For the convenience of the reviewer we report the image here. The image is a post-surgical echography of the implanted POLYRETINA (indicated by white arrows) showing its tight apposition to the retina. It is visible that the implant really touches the retina everywhere.

We should note that during the surgery (see **Supplementary Video 1**) we removed the lens (which cannot be replaced in the minipig). Therefore, echography is the best/only way we have to image the implant. Also, to complement this analysis, we provide evidence that different points of the implant can activate the retina (**Fig. 8**). These results indicate that the implant touches the retina on its entire surface.

Pressure is a different topic and an interesting one. To the best of our knowledge, it is not possible to measure the pressure exerted by an implant on the retina in-vivo. However, pressure and long-term mechanical stability are valid points. We already discussed mechanical stability in the discussion section, and we now added pressure [page 7, “POLYRETINA is held in position and fixed to the sclera using two retinal tacks, as it is conventionally conducted with epiretinal prostheses. Although retinal tacks have been used in surgical practice for many years, their use for retinal prostheses raises some concerns. In particular, the long-term mechanical stability of the implant and its anchoring to the retina might be affected. Also, it is difficult to control the pressure of the retina during surgery and in the follow up period. Therefore, improvements in the fixation method are desirable.”].

I have great difficulty assessing the number of animals in each treatment arm. It is not absolutely clearly stated how many animals served where. Was it really only one animal in the establishing degeneration trial? Was there any power analyses performed? Why not give a table where the reader can follow what was performed in which animal with which results? There seems to be a lack in rigorous analysis. Or in will of publishing it. The task is gerat now doubt, but it would easier to judge it even higher when we get the feeling that the authors have set very high standards for themselves by publishing all the data (not merely examples), by self-critically examining the results. No need to fear, the accomplishments are great!

Re: we have included a table summarising the animals used in the study (see **Tab. 1** at page 9). We hope this table will help the reviewer to identify which animal was used where.

We would like to specify that the number of eyes and animals used in each test was already indicated in every analysis, usually with the format: n = X eye from N = Y minipigs. There is no intention here to hide data and the statistical analysis is as rigorous as it can be. Power analysis was performed as reported in our answer to Reviewers 1-2 in the previous round of revision. It is stated in the manuscript [Page 5, “**Sample size was determined with a priori power analysis (alpha = 0.05, power 0.85).**”]. Moreover, the achieved power for our statistical analysis is reported for every test [see for example at page 6, “The statistical analysis revealed that peak-to-peak amplitudes are significantly higher after implantation compared to before implantation (p = 0.0074, power = 1.00; two-tailed paired t-test)”. All tests reported in the study showed that the planned number of animals is sufficient to achieve high statistical results with high power (power of 0.98 or higher). Therefore, we believe that the number of animals is adequate from a statistical point of view, and use of new animals cannot be justified.

The reviewer is concerned that only one animal was used for the degeneration study. This is not the case. In **Tab. 1** we reported that 8 animals were used for that part of the study. For the histological analysis, animals had to be split among the different time points and type of stainings (H&E vs IHC), so one animal was used for each time point.

Lastly, we would like to clarify that we do not show ‘merely examples’. All data are presented with examples and cumulative statistics for all the animals. For example, for the OCT data, **Fig. 1**, **Supplementary Fig. 1**, and **Supplementary Fig. 2** show representative examples and the full statistical analysis across animals is in **Supplementary Fig. 3**. For the ERGs data, **Fig. 2a** and **2d** show examples and the other panels in the figures show cumulative data with statistics. In **Fig. 7**, panels **d** and **e** are representative examples and the other panels show cumulative analysis and statistics. The same is valid for all the other data.

There is a lack of discussing criticcally the results. Strengths? Weaknesses (honestly, what lacks?, what misses? What failed? What will be the difficulties in the future? In humans? Costs? Number of animals?). I get the impression that the authors fear a rejection if this all would be critically illuminated. But that is (at least for me) not the case. I want the truths, and the harder the authors are with themselves, the more I trust them. Instead the authors lose themselves in bloomy descriptions and difficult sentences.

Re: we expanded the discussion to highlight some of the elements indicated by the reviewer. We would like to point out that some aspects cannot be discussed here. For example, cost is something not related to the research effort but to the business aspects. Also, we would like to clarify that we included a critical discussion point in every paragraph of our discussion.

For instance, about the animal model, we wrote: “On the other hand, IAA-treated minipigs do not recapitulate the complex remodelling processes undergoing in the retinas of patients affected by retinitis pigmentosa, which might hinder the translation of these results to patients”.

About mechanical stability, we wrote: “POLYRETINA is held in position and fixed to the sclera using two retinal tacks, as it is conventionally conducted with epiretinal prostheses. Although retinal tacks have been used in surgical practice for many years, their use for retinal prostheses raises some concerns. In particular, **the long-term mechanical stability of the implant and its anchoring to the retina might be affected. Also, it is difficult to control the pressure of the retina during surgery and in the follow up period.** Therefore, improvements in the fixation method are desirable.”

About the functional recovery, we wrote: “However, it is important to note that perception of light is just one step towards artificial vision. A key question is about spatial resolution, which was not assessed in this work

due to the challenges in recording cortical evoked potentials in minipigs. However, in a previous study performed ex-vivo with retinal explants from blind mice, we reported that POLYRETINA holds a stimulation resolution equivalent to its pixel pitch, which is 120 μm ".

Finally, we now added the following sentence about further studies: "Further studies will be required to assess the device in the long-term, addressing the functional and mechanical stability of the implant and its biocompatibility".

These points mentioned in the discussion recapitulate the most important scientific elements that must be considered for the clinical translation.

Reviewer Comments, third round -

Reviewer #1 (Remarks to the Author):

The authors now provide convincing evidence of electrically evoked cortical responses after implantation of the device.

I have still one concern about figure 8 which does not demonstrate convincing evidence of focal responses. 1) The signals are averaged over 20 repetitions as opposed to 200 repetitions for the wider field stimulation, 2) the shape of the fEEP does not resemble the responses from fig 7 and, 3) there is no control condition.

In my opinion, this figure and the associated claim should be removed from the paper before the manuscript can be accepted.

Otherwise very nice paper and congratulations for all the work.

Reviewer #2 (Remarks to the Author):

The authors have made extensive efforts to address reviewer concerns. Although I am still concerned about the IAA model that the reviewers have chosen. It is a blunt force inhibition of glycolysis aimed at GAPHD. Rods die from this, but cones survive. They lose OS and function for some period up to 3 weeks to a month, then they regain function. That was/is the concern regarding a photopic flash with this model. But I think that the authors have done what they can to address these concerns.

Reviewer #3 (Remarks to the Author):

Good, work. Now it is well presented. No objections to publications.

Replies to the comments are in blue.

Reviewer #1 (Remarks to the Author):

The authors now provide convincing evidence of electrically evoked cortical responses after implantation of the device.

I have still one concern about figure 8 which does not demonstrate convincing evidence of focal responses.

1) The signals are averaged over 20 repetitions as opposed to 200 repetitions for the wider field stimulation, 2) the shape of the fEEP does not resemble the responses from fig 7 and, 3) there is no control condition.

In my opinion, this figure and the associated claim should be removed from the paper before the manuscript can be accepted. Otherwise very nice paper and congratulations for all the work.

Re: we removed Figure 8 and the associated claims. We thank the reviewer for the positive assessment.

Reviewer #2 (Remarks to the Author):

The authors have made extensive efforts to address reviewer concerns. Although I am still concerned about the IAA model that the reviewers have chosen. It is a blunt force inhibition of glycolysis aimed at GAPDH. Rods die from this, but cones survive. They lose OS and function for some period up to 3 weeks to a month, then they regain function. That was/is the concern regarding a photopic flash with this model. But I think that the authors have done what they can to address these concerns.

Re: we thank the reviewer for the positive assessment. We have discussed the issue with the model in the discussion, as follows: "Also, IAA is a GAPDH (Glyceraldehyde-3-Phosphate Dehydrogenase) inhibitor leading to rod degeneration and cone inactivation. Previous results showed that cone responses are rescued 5 to 6 weeks after IAA administration³⁹. However, in our study light-adapted fERGs were suppressed for the entire testing period (up to 11 weeks after IAA administration) indicating no recovery of cone responses. For POLYRETINA validation, we repeated IAA administration twice and compared the responses to light immediately before and after POLYRETINA implantation. Therefore, we exclude any possible contribution from remaining or recovered photoreceptors". In addition, this specific point was already mentioned in the results section, as follow: "A previous report about IAA-treated pigs showed recovery of the cone response 5-6 weeks after IAA administration³⁹. To rule out this possibility in the Göttingen minipig model, we performed a longitudinal study in both IAA-treated and untreated minipigs (Fig. 2g). Light-adapted fERGs showed suppression of the a-wave and b-wave in IAA-treated minipigs (n = 7 eyes from N = 5 minipigs) and no recovery for the entire testing period. On the other hand, they remained stable in untreated minipigs (n = 4 eyes from N = 2 minipigs).".

Reviewer #3 (Remarks to the Author):

Good, work. Now it is well presented. No objections to publications.

Re: we thank the reviewer for the positive assessment.